

# A High Speed Particle Phase Discriminator (PPD-HS) for the classification of airborne particles, as tested in a continuous flow diffusion chamber

Fabian Mahrt[1], Jörg Wieder[1], Remo Dietlicher[1,*], Helen R. Smith[2], Chris Stopford[2], and Zamin A. Kanji[1]

[1]Institute for Atmospheric and Climate Science, ETH Zurich, 8092 Zurich, Switzerland
[2]Centre for Atmospheric and Climate Physics, University of Hertfordshire, Hatfield, Hertfordshire, AL10 9AB, United Kingdom
*now at: Center for Climate System Modeling, ETH Zurich, 8092 Zurich, Switzerland

*Correspondence to:* F. Mahrt (fabian.mahrt@env.ethz.ch) and Z. A. Kanji (zamin.kanji@env.ethz.ch)

**Abstract.** A new instrument, the High Speed Particle Phase Discriminator (PPD-HS) developed at the University of Hertfordshire, for sizing individual cloud hydrometeors and determining their phase is described herein. PPD-HS performs an in-situ analysis of the spatial intensity distribution of near forward scattered light for individual hydrometeors yielding shape properties. Discrimination of spherical and aspherical particles is based on an analysis of the symmetry of the recorded scattering patterns. Scattering patterns are collected onto two linear detector arrays, reducing the complete 2D scattering pattern to scattered light intensities captured onto two linear, one dimensional strips of light sensitive pixels. Using this reduced scattering information, we calculate symmetry indicators that are used for particle shape and ultimately phase analysis. This reduction of information allows for detection rates of a few hundred particles per second.

Here, we present a comprehensive analysis of instrument performance using both spherical and aspherical particles, generated in a well-controlled laboratory setting using a Vibrating Orifice Aerosol Generator (VOAG) and covering a size range of approximately $3 - 32$ µm. We use supervised machine learning to train a random forest model on the VOAG data sets that can be used to classify any particles detected by PPD-HS. Classification results show that the PPD-HS can successfully discriminate between spherical and aspherical particles, with misclassification below 5 % for diameters $> 3$ µm. This phase discrimination method is subsequently applied to classify simulated cloud particles produced in a continuous flow diffusion chamber setup. We report observations of small, near-spherical ice crystals at early stages of the ice nucleation experiments, where shape analysis fails to correctly determine the particle phase. Nevertheless, in case of simultaneous presence of cloud droplets and ice crystals, the introduced particle shape indicators allow for a clear distinction between these two classes independent of optical particle size. We conclude that PPD-HS constitutes a powerful new instrument to size and discriminate phase of cloud hydrometeors and thus study microphysical properties of mixed-phase clouds, that represent a major source of uncertainty in aerosol indirect effect for future climate projections.



# 1   Introduction

Microphysical processes in clouds involving ice particles contribute to major uncertainties in cloud formation, evolution and precipitation formation (Mülmenstädt et al., 2015) and subsequently to radiative properties associated with these clouds on both regional and global scales (Matus and L'Ecuyer, 2017; McCoy et al., 2016; Tan et al., 2016). A major limitation remains the

accurate phase partitioning between water and ice particles which requires reliable discrimination of supercooled liquid cloud droplets from ice crystals in mixed-phase clouds (MPCs). For instance, for the same water content, ice clouds are optically thinner compared to liquid water clouds and thus reflect less shortwave radiation back to space (Lohmann, 2017), whilst simultaneously trapping more longwave radiation in the Earth atmosphere system, due to their lower cloud top temperatures. Recently, Korolev et al. (2017) reviewed MPCs and concluded that a major limitation in studying these clouds is associated

with the availability of instrumentation that is able to distinguish cloud droplets from ice crystals. For particles below 100 µm (diameter) reliable measurements are especially scarce (Baumgardner et al., 2017). The operating principle of many cloud probe instruments, such as the Cloud Droplet Probe (CDP, Lance et al., 2010) and the Forward Scattering Spectrometer Probe (FSSP, Jaenicke and Hanusch, 1993), relies on a Mie approximation (Mie, 1908), relating the scattered light intensity (integrated over a given angular range) to size for spherical particles. The particle size can theoretically be measured if the

wavelength of the incident light and refractive index of the particles under consideration are both known (Baumgardner et al., 2011). Such instruments are widely used for sizing and counting individual particles, but usually do not offer techniques to determine cloud particle phase. Particle phase is thus often segregated based on particle size alone, whereby small, near-spherical particles are assumed to be liquid. This is despite the knowledge that cloud particle size distributions comprised of a mixture of cloud droplets and ice crystals are affected by the presence of small ice particles. Thus, a distinction of ice and

water particles purely by optical size is precarious (Heymsfield et al., 2006) and can lead to an overestimation of the number concentration of cloud droplets when both solid and liquid phases are present in the small size channels (Gardiner and Hallett, 1985).

Nevertheless, particle size reported by optical scattering instruments is often used to infer particle phase during continuous flow diffusion chamber (CFDC) studies (e.g. Rogers, 1988). This approach is usually justified by defining (operational) optical

size thresholds above which liquid droplets are not expected to be present because of their slower diffusional growth compared to ice crystals. Additionally, passing all hydrometeors formed within a CFDC through a so-called evaporation section, where the relative humidity ($RH$) is held below water saturation, promotes the evaporation of cloud droplets upstream of any optical detection unit, whilst preserving the ice crystals for detection. However, the transit time and temperature in the evaporation section limits the dynamic $RH$ conditions at which ice crystals can be reliably discriminated from water droplets. In some

experiments, no evaporation section is used and the dynamic $RH$ range for using particle size to distinguish droplets from ice crystals is reduced further compared to methods that deploy an evaporation section.

To overcome such limitations, scattering probes with phase-discriminating capabilities have been deployed. A common way to determine hydrometeor phase in CFDC studies encompasses polarization analysis (e.g. Zenker et al., 2017; Nicolet et al., 2007), making use of the fact that aspherical particles change the polarization of incident light, whereas spherical



particles do not. In the cases where small near-spherical ice crystals can form, depolarization techniques might be limited in the discrimination between spherical liquid drops and ice crystals. Other particle properties, however, can also be used to estimate particle phase. Particle shape, for instance, constitutes a powerful parameter that can be used to discriminate hydrometeor types (Hirst and Kaye, 1996).

The spatial intensity distribution of light scattering events, as for example 2D forward scattering patterns, exhibit various features unique to the scatterer. The infinite rotational symmetry of a perfect sphere for instance, will result in a scattering pattern comprising of a rotationally symmetric set of diffraction fringes, where the fringe spacing is determined by the droplet size. Ice crystals on the contrary are usually associated with basal and prism faces of different extents (Libbrecht, 2005; Lohmann et al., 2016) and small-scale complexity such as surface roughness (e.g. Voigtländer et al., 2018; Magee et al., 2014;

Järvinen et al., 2018), resulting in optically anisotropic scattering patterns. In case an ice crystals is associated with a hexagonal crystal structure, forming a six-fold symmetry about one axis (the basal face), the resulting diffraction pattern will exhibit a six-fold symmetry, in case these ice crystals are oriented in such a way that the z-axis is aligned with the optical axis of the image laser of PPD-HS (see Supplementary Information (SI) Fig. S1). In reality, however, the 2D forward scattering patterns do not always exhibit perfect symmetry. For example, liquid particles may become oblate in air flows, and the scattering patterns from

crystalline materials are highly dependent upon particle orientation and can be modified by the presence of surface roughness (Järvinen et al., 2016, 2018). Despite these complexities, the symmetry of the scattering pattern can be used to infer various particle properties.

      Over the past two decades, many instruments analyzing spatially resolved scattering profiles have been built for purposes of cloud particle detection (Baumgardner et al., 2011, and references therein). Recently, Vochezer et al. (2016) described the

PPD-2K instrument (Particle Phase Discriminator mark 2, Karlsruhe edition;  Kaye et al., 2008), a laboratory edition of the Small Ice Detector (SID-3, Ulanowski et al., 2014), which records high resolution scattering patterns using an intensified Charge-Coupled Device (CCD) camera, as shown by the examples depicted below.

A major limitation of these devices is the low frame rate on the order of a few tens of particles per second, that limits the number of single particles sampled. This becomes problematic when hydrometeors of different phase are spatially inhomogeneously

distributed within clouds (Korolev and Isaac, 2006; Beck et al., 2017). Besides, high frames rates are beneficial for laboratory studies, where particle number concentration easily exceed a few tens of particles per cubic centimeter, and low frame rates lead to coincidence errors, even at moderate number concentrations (Cotton et al., 2010).

      In order to overcome the low frame rates of previous optical devices, Stopford et al. (2013) deployed linear Complementary Metal-Oxide-Semiconductor (CMOS) arrays rather than a CCD camera, when investigating the light scattering of asbestos

fibres. The reduction in light scattering (pixel) information captured by two linear CMOS array as compared to the complete 2D scattering pattern recorded by a CCD camera allows for significantly increased detection rates (El-Desouki et al., 2009), as less information has to be processed and stored. Recent advances in data analysis techniques available within the field of machine learning (ML) provide powerful tools to handle large data sets. Rather than following a rule-based scheme to split particles within a data set, ML techniques determine the parameters (called predictors), which categorize the data best. Hence,



these methods provide enormous potential to facilitate the particle classification problem faced in MPC studies and with that push our understanding of the microphysical processes therein.

Here, we present a new instrument called the High Speed Particle Phase Discriminator (PPD-HS), to discriminate cloud hydrometeor phase. Specifically, PPD-HS developed by the University of Hertfordshire, UK, is designed to capture the spatial

intensity distribution of forward scattered light by airborne particles on two linear CMOS arrays on a Particle by Particle (PbP) basis. Since the resulting scattering pattern is a function of particle size, shape, and orientation with respect to the incident light as well as the polarization and wavelength of the incident light (Hirst and Kaye, 1996), some morphological features of the particle can be inferred (Ulanowski et al., 2012), with size and sphericity being of interest to this study. To assess the performance of PPD-HS, characterization experiments have been carried out using laboratory generated particles of known and

well-defined size and geometry. Using this calibration data set, a random forest model is trained to classify particles detected by PPD-HS. Finally, we examine the discrimination of simulated cloud hydrometeors as either ice crystals or cloud droplets from a set of experiments when PPD-HS was coupled to the Horizontal Ice Nucleation Chamber (HINC, Lacher et al., 2017; Mahrt et al., 2018), operated at various conditions relevant for cirrus, liquid and MPCs using different aerosol species, where the phase of the formed particles in HINC is thermodynamically predictable.

## 15   2   Description of PPD-HS

### 2.1   Overview and flow configuration

Particles are drawn into the instrument by an external pump through the inlet, a 6 mm inner diameter stainless steel pipe, tapered to 2 mm over a length of 45 mm. The resulting laminar flow has a parabolic flow-profile, causing elongated particles such as fibres, columns and needle shaped ice crystals to be preferentially aligned along the flow axis (Lin et al., 2004;

Abdelmonem et al., 2011, see SI Fig. S3). The inlet is mounted on the optics block via a Klein Flange fitting, threaded into the optics block and sealed by an O-ring. In addition to the sample flow, $F_p$, a filtered bleed flow, $F_{bleed}$, purges the scattering chamber of particles that escape the sample flow, and prevents particle deposition on the optical surfaces. The sum of $F_p$ and $F_{bleed}$ forms the total flow rate through PPD-HS, $F_{tot}$, which can be varied between approximately 2 Lmin$^{-1}$ and 10 Lmin$^{-1}$ (see SI Fig. S4).

### 25   2.2   Particle detection and sizing

Inside the scattering chamber, shown in Fig. 1a, particles are exposed to two laser beams, as illustrated in Fig. 1b. Firstly, particles pass a 35 mW continuous wave trigger laser beam (658 nm diode laser, Optoelectronics Inc., part # HL6501MG), which is used for particle detection and sizing. The beam is apertured to transform the initial Gaussian intensity profile into a top-hat profile. The beam is then expanded to a width of 4.5 mm in the plane normal to the sample flow, and focused at the

intersection of the particle flow and the trigger laser to a depth of $\sim 0.1$ mm. As the sample flow, which has a diameter of approximately 2 mm at the position of the trigger laser, lies wholly within the trigger laser beam, the so-called sensing-volume



of the trigger laser (gray dot on top of the trigger laser beam in Fig. 1a) is defined mechanically by the flow speed and inlet size. Scattered light between approximately 10.6 ° and 101.0 ° to the laser beam axis (light orange shading) is collected by a spherical mirror (M1, Edmund Optics, part #43467), and focused onto a silicon photodiode (Edmund Optics, part #54-035 ). Over this angular range, dominated by sideways scattered light, a near monotonic relation between photodiode intensity and particle size is achieved.

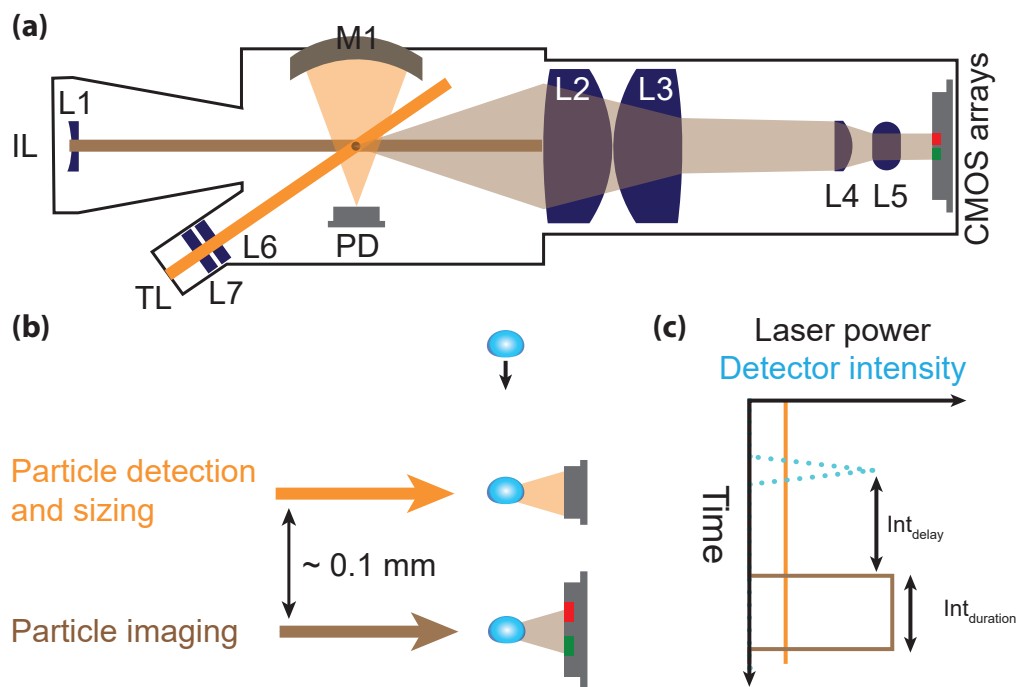

**Figure 1.** Schematic of PPD-HS along with an illustration of its working principle. (a) Detailed top view of the optics, (b) simplified side view illustrating vertical displacement of the lasers used for particle detection and sizing as well imaging, and (c) corresponding signal and laser switch. L1 to L7 denote optical lenses, M1 a parabolic mirror, IL imaging laser, TL trigger laser and PD photodiode. Light orange and brown shading in (a) and (b) correspond to scattered light of trigger laser beam and image laser beam, respectively.

## 2.3 Particle imaging

The intensity signal recorded by the photodiode is used to assess particle size, and whether the particle is central to the sample flow. If the intensity of the light scatter meets a user-defined threshold, a trigger signal is sent to the image laser to initiate the subsequent scattering pattern acquisition process. The image laser is vertically displaced from the trigger laser by

10    approximately 0.1 mm (Fig. 1b). The image laser is a pulsed ≈ 250 mW, 638 nm multimode diode laser (Optnext Japan Inc., part HL6388MG). The beam is focused at the intersection with the sample flow, and apertured twice before reaching the scattering volume. The first aperture limits the width of the beam to approximately 3 mm in order to reduce stray light and



evens the energy distribution across the beam to limit the variation to approximately 25 %, minimizing erroneous classification of particles with trajectories close to the beam edge. The second aperture, in front of the scattering volume, further reduces stray light. Laser firing and duration times are adjustable by the user through the trigger delay and integration time (Fig. 1c) in order to account for the different total flow rates at which PPD-HS can be operated. For particles illuminated by the image laser,

light scattered in the forward direction between $\sim 6\,°$ and $20\,°$ is collected and passed through an optical assembly composed of a series of lenses (L2-L5) before impinging on the detector arrays (light brown shading, Fig. 1a). The lens assembly is designed such that a beam dump, mounted around the center axis of the image laser beam on the surface of L2, behind the sensitive volume, absorbs the direct laser beam. The use of two cylindrical lenses L2 and L3 allows to vertically compress the scattering pattern and thus increases the elevation angle of scattered light detected by the CMOS arrays. The detector arrays

are composed of two linear CMOS arrays (Hamamatsu Corp., Japan, Model S9227) aligned vertically and spaced laterally symmetrically $4.2\,\mathrm{mm}$ away from the center of the optical axis. CMOS arrays are chosen to exhibit an almost linear response over a wide dynamic size range in order to cover a particle size range between $\approx 2\,\mathrm{\mu m}$ and $\approx 300\,\mathrm{\mu m}$. At the same time, CMOS arrays have both high read-out speed, allowing for high particle detection rates, and small pixel size, producing high resolution scattering patterns, with each array being composed of $512$ vertically aligned, individually sensed pixels. The intensity signal

recorded by each pixel is integrated during the activation of the image laser (integration time, Fig. 1c) and corrected by a background value (see SI Sect. S5.1).

The example in Fig. 2a illustrates an interference pattern that is radially symmetric around the center line of the optical axis, comprised of concentric intensity maxima (white rings), obtained from spherical particles, such as liquid cloud droplets. The scattered light, falling on the detector plane indicated by the squares on top of Fig. 2a, defines a two-dimensional transformation

of the three-dimensional intensity distribution of the light scattered by a particle in the near forward direction. Furthermore, the light intensities recorded by the two linear detector arrays denote one dimensional strips out of the complete two-dimensional scattering image (red and green colored pixels). In the case of a spherical particle the light scattering information captured by the detectors of PPD-HS is given by the intersection of the CMOS arrays with the diffraction fringes, resulting in a scattering pattern, as illustrated in Fig. 2d, where the vertical dimension covers the 512 pixels and the horizontal axis indicates relative

forward light scattering intensity for the two CMOS arrays, shown in different colours for clarity. Symmetry evaluation of these scattering patterns is subsequently used to determine particle shape, with spherical and aspherical particles producing symmetric and asymmetric scattering patterns, respectively (see Sect. 2.5). As particle phase is ultimately related to particle shape, we constrain our discussion to particle shape in the following.

All data on optical particle size and scattering patterns are collected for individual particles and thus allow to quantify optical

properties on a PbP basis.

## 2.4 Electronic data acquisition configuration

There are two different sets of electronics reading out the CMOS data. One to perform rapid analysis of the CMOS data, providing real-time feedback to the user containing information of different scattering pattern parameters along with particle size information and information of the instrument settings. The other electronics board collects and stores the raw data, i.e.



the intensity values for the individual pixels of the detector arrays, for subsequent offline processing and analysis. Both boards are based around a field programmable gate array. Reduction of scattering information along with implementation of fast electronics, is key for the high detection rate of PPD-HS (see SI Tab. S1), with CMOS dead time being the rate limiting factor in case of PPD-HS (see SI Sect. S5.3).

The readout of the real time-electronics is triggered by an analogue circuit detecting an intensity peak from the photodiode. It stores this peak value for later conversion to particle size, and controls the pulsing of the image laser, as described above. When the real time-electronics begins readout of the CMOS arrays, it signals the raw-electronics to readout simultaneously, which is not connected to the photodiode and thus its output does not contain any information on particle size (see SI Sect. S4.1). If either set of electronics is still processing when another particle arrives, that board will not readout that trigger. This

means that particle information of the two electronics boards are complementary but not congruent.

## 2.5 Phase discrimination indicators

Figures 2d-f show examples of CMOS array data for individual airborne particles of different shapes. As can be seen from Fig. 2a, the rotational symmetry of a spherical particle translates into a nearly perfect azimuthal symmetry in the 2D scattering pattern, i.e. nearly perfect alignment of the intensity peaks across the two arrays as well as symmetry around the center line of

each array. On the other hand, the scattering pattern of a more complex-shaped, aspherical particle is a jagged pattern, showing multiple randomly arranged peaks with overall little symmetry of these peaks within one array and between the two arrays (Fig. 2b). Finally, needles and/or columnar particles are characterized by very distinct scattering patterns comprised of a single sharp peak along each array, as shown in Fig. 2c. Here, fibre-like particles were used to allow for discrimination between columnar ice crystals and (hexagonal) plates, both constituting an aspherical particle class. For particle shape evaluation and

associated phase classification, a combination of different indicators is calculated from these scattering patterns, as described in the following.

Similarly to Stopford et al. (2013) we calculate a peak-to-mean (PTM) ratio given as:

$$PTM = \frac{I_{\mathrm{max,x}}}{\bar{I}_{\mathrm{x}}}, \tag{1}$$

where, $I_{\mathrm{max,x}}$ denotes the maximum intensity recorded by a pixel on array $x$ and $\bar{I}_{\mathrm{x}}$ the mean scattering intensity along that

array. Fibre-like particles, characterized by an intense and narrow peak on both arrays (Fig. 2c), will yield relatively large PTM values, whereas values for spheres and aspherical (non fibre-like) particles will be significantly lower as their intensity patterns cover a larger pixel range. It should be noted that an asphericity factor as e.g. used by Hirst et al. (2001) and Zhang et al. (2016) cannot be calculated, due to the linearity of the detector arrays in PPD-HS. Hirst et al. (2001) calculate an asphericity factor ($A_{\mathrm{f}}$), as a measure of variation of scattered light intensity detected by the pixels across their *circular* detector array. For highly

symmetrical scatterers, i.e. spherical particles, producing ring-shaped interference patterns, $A_{\mathrm{f}}$ should theoretically yield values of zero. More aspherical particles yield larger values.

For our linear detector system, we developed a new shape indicator, called top to bottom comparison (TBC), that can be used to investigate the symmetry of a scattering pattern along one array. TBC sums the absolute differences of equidistant pairs of

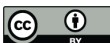



**Figure 2.** Top row: Example 2D scattering patterns captured by SID3 (Ulanowski et al., 2014), showing (a) a cloud droplet, (b) an ice crystal and (c) a columnar ice crystal. The forward scattering pattern of the cloud droplet reveals rotational symmetry, while those of the ice crystal and the fibre do not. The pixels of the CCD camera are schematically indicated by the gray squares on top of panel (a). The reduced information captured by the linear CMOS arrays of PPD-HS are highlighted by the red (array 1) and green (array 2) squares, respectively. Bottom row: Panels (d)-(f) show scattering data from the linear CMOS arrays, showing relative forward scattered light intensity for the individual pixels comprising an array, for (d) a spherical particle, (e) an aspherical particle and (f) a fibre-like particle. Images a-c provided by C. Stopford.

pixels around a midpoint pixel, normalized by the maximum intensity. Mathematically the TBC is defined as:

$$TBC_x = \frac{1}{2 \cdot p_{max} \cdot I_{max,x}} \sum_{p=0}^{p_{max}} |(I_{cx+1+p,x} - I_{cx-p,x})|, \qquad (2)$$

where the subscript "$cx$" denotes the midpoint pixel of a CMOS array $x$ and $p$ the pixel number relative to this midpoint pixel. The number of pixels relative to the midpoint that are considered for the TBC calculation is given by $p_{max} = min(cx -$





$1, 511 - cx$). Perfectly spherical scatterers would also yield TBC values of zero (neglecting electrical noise), whereas aspherical particles such as ice crystals yield larger TBC values. Normalization to the maximum intensity is needed to compare the TBC of hydrometeors of different (physical) sizes, which produce light scattering patterns of different absolute intensities (see for instance intensity values in Fig. 2d-f). At the same time, normalization to the total number of pixels considered for TBC

calculation ($2 \cdot p_{\text{max}}$) ensures comparability of TBC values with different midpoint configurations, i.e. pixel information content.

Similarly to the TBC, the scattering patterns captured by each CMOS array can also be directly compared. We therefore calculate the so-called array inter-comparison (AIC) indicator:

$$AIC = \frac{1}{2 \cdot p_{\text{max}}} \left[ \sum_{\text{p}=0}^{p_{\text{max}}} \left| \frac{I_{\text{c1}+1+\text{p},1}}{I_{\text{max},1}} - \frac{I_{\text{c2}+1+\text{p},2}}{I_{\text{max},2}} \right| + \sum_{\text{p}=0}^{p_{\text{max}}} \left| \frac{I_{\text{c1}-\text{p},1}}{I_{\text{max},1}} - \frac{I_{\text{c2}-\text{p},2}}{I_{\text{max},2}} \right| \right], \tag{3}$$

Again, an equal number of pixels is compared around the midpoint pixels, but also across the two CMOS arrays. The total

number of pixels considered for the calculation of AIC is thus given by $p_{\text{max}} = min(c1 - 1, 511 - c1, c2 - 1, 511 - c2)$. For spherical particles, the relative intensities captured by both arrays should theoretically cancel out, whereas aspherical particles, causing the intensity peaks to be randomly distributed along the CMOS arrays, yield larger values.

It should be noted that the midpoints are not necessarily equivalent to the physical center (pixel = 256) of the CMOS array, but were found to be 262 and 258 for array 1 and array 2, respectively (see Appendix A).

## 15  3  Methods

### 3.1  Experimental setup

#### 3.1.1  PPD-HS calibration measurements

Different types of particles were used for PPD-HS calibration experiments. A Vibrating Orifice Aerosol Generator (VOAG, TSI Inc., Model 3450) was used to produce almost monodisperse populations of both spherical and aspherical particles as

proxies for cloud droplets and ice crystals, respectively (see SI Fig. S12). The experimental setup is shown in Fig. 3. Within VOAG, moving a solution with a constant flow rate through a micrometer-sized metal orifice, results in a constant cylindrical liquid jet ejected from the orifice. The assembly holding the orifice vibrates at a constant frequency, which periodically breaks up the liquid jet, resulting in solution droplets of uniform mass. In an evaporation column downstream of the VOAG, the solvent evaporates, resulting in equally sized solute particles. In order to produce spherical particles, solutions of 2-propanol

and polyethylene glycol (PEG-400, Sigma Aldrich, BioUltra 400, hereafter referred to as PEG) were used. The low vapor pressure of PEG at room temperature ($< 0.01$ mmHg) allows for production of uniform droplet sizes. PEG has a refractive index of $1.466$ at $589$ nm (Ottani et al., 2002) and we assume a shape factor of unity. For the production of aspherical particles, mixtures of 2-propanol, Milli-Q water and NaCl were used, resulting in formation of solid salt crystals upon evaporation of the solvents. The VOAG was operated with a $20$ µm and $35$ µm diameter orifice, a disperison air flow rate of $1.5$ Lmin$^{-1}$ and

a dilution air flow rate of $60$ Lmin$^{-1}$, using particle-free compressed air. Adjustment of the vibration frequency and solute



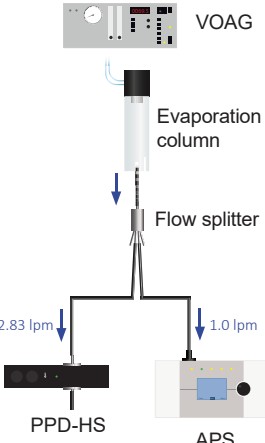

**Figure 3.** Experimental set up used for PPD-HS calibration measurements. A VOAG was used to generate both spherical and aspherical particles of different sizes. An APS was operated in parallel to PPD-HS, in order to monitor the particle size distribution (see example size distribution in SI Fig. S12). A vacuum pump, attached downstream of PPD-HS, is used to draw air through PPD-HS.

to solvent ratio allows to produce different particle sizes. Here, we generated particles with diameters between approximately $3 - 32 \ \mu m$ covering the typical size range of freshly formed cloud particles, where phase segregation remains an ongoing issue. At the same time it covers the size range of hydrometeors typically formed within CFDCs (up to $\approx 10 \ \mu m$, Rogers, 1988). For each size, between approximately $1,000$ and $400,0000$ single particles were sampled by PPD-HS (see SI Tab. S1).

As depicted in Fig. 3 we inverted the droplet generator assembly of the VOAG on top of the vertically aligned evaporation column to allow for sampling of supermicron particles, minimizing their gravitational loss within the setup. A flow splitter (TSI Inc., Model 3708) was mounted at the bottom of the drying column with the inlet extending $\approx 3$ cm into the column. The flow is split to an Aerodynamic Particle Sizer (APS, TSI Inc., Model 3321; $1.0 \ \mathrm{Lmin^{-1}}$) and PPD-HS ($2.83 \ \mathrm{Lmin^{-1}}$), using $6$ mm conductive tubing. The flow through PPD-HS is equal to $F_{tot}$ and is accomplished using a vacuum pump, which

is controlled using a needle valve. Flows were regularly checked at the beginning and the end of an experiment and variation was found to be $< 5 \ \%$. The APS was used to monitor particle size distribution and number concentration. The evaporation column was not capped at the bottom, to allow the exhaust air to escape and avoid turbulence. Finally, fibre-like particles were wet-generated from a suspension of coal burning sourced fly ash particles and Milli-Q water, using an atomizer, as described in more detail in Grawe et al. (2016, 2018).

**3.1.2   PPD-HS coupled to HINC**

A series of cloud chamber experiments were performed to simulate liquid, MPC and cirrus cloud conditions. Generation of ice particles and cloud droplets is achieved by means of HINC, operated upstream of PPD-HS. HINC is a CFDC and has recently been described in detail elsewhere (Lacher et al., 2017; Mahrt et al., 2018). Experiments using HINC were conducted using





two different aerosol species. In a first set of experiments $NH_4NO_3$ particles were used, atomized from a 0.1 M solution and size selected for a mobility diameter of $d_m = 100$ nm using a Differential Mobility Analyzer (DMA, TSI Inc., classifier Model 3080, with a 3081 column and a polonium radiation source) and an aerosol to sheath flow ratio of 1:10. $NH_4NO_3$ aerosol was used in order to produce cloud droplets at $T > 235$ K and $RH_w > 100$ %, as well as ice crystals at $T \leq 235$ K where

homogeneous freezing rates are expected to cause significant freezing (Koop et al., 2000). These experiments provide ice-only and droplet-only cases in order to cross-check the shape discrimination capability of PPD-HS.

In a second set of HINC experiments we used illite NX (Arginotec, NX, Nanopowder), a mineral dust aerosol, dry suspended from a Fluidized Bed Aerosol Generator (FBAG, TSI Inc., Model 3400A) and DMA size selected to $d_m = 400$ nm, operated at an aerosol to sheath flow ratio of 1:7. Illite NX is widely used in ice nucleation experiments (e.g. Welti et al., 2009; Hiranuma

et al., 2015) as proxy for atmospheric dust and is expected to form a mixture of cloud droplets and ice crystals at $T > 235$ K and $RH_w > 100$ %, providing us with a test case for PPD-HS to discriminate the shape in the presence of both crystals and droplets, as is the case in MPCs.

## 3.2 Particle shape classification: supervised machine learning

Particle shape is ultimately inferred from classification of the PbP data using a supervised ML approach. Supervised ML is

frequently used for laboratory collected data, sampled under controlled conditions, where correct classification outcomes are known through the production process of the particles. In general, supervised ML algorithms are used to assign new, unknown data to predetermined classes through similarity analysis of the new data and the data comprising the a priori determined target classes. In the case of PPD-HS, the target classes are spherical particles (cloud droplets) and aspherical particles (ice crystal). Thus, these target classes require training data that cover the given classes (Mohri et al., 2018). We have generated calibration

data sets for each of these classes using the VOAG experimental set up as described in Sect. 3.1.1.

The supervised ML approach used to classify the PPD-HS PbP data is a random forest model, based on MATLAB's (Math-Works Inc., R 2018b) *TreeBagger* algorithm. A random forest model (Breiman, 1996, 2001) constitutes an ensemble of decision trees that are combined through averaging over multiple individual trees. Decision tree approaches have previously been used for cloud particle classification specifically, e.g., Garimella et al. (2016); Bernauer et al. (2016), and are frequently used for

any classification problems, such as data from single particle mass spectrometry (e.g. Christopoulos et al., 2018). Ruske et al. (2018) recently identified best performance of random forest models when comparing different ML approaches to classify different particle types.

In an individual decision tree, a matrix of feature vectors encompassing the PbP data is used and the tree is formed through consideration of all possible splits across *all* predictors (variables), ultimately choosing the predictor that divides the data best,

through maximization of particle class separation within the entire phase space. However, individual decision trees have the tendency to over fit the data and are thus often too specific to the training data used, i.e. they do not cover the entirety of particle types that might be collected during conditions not encountered in the training data set, with the latter being a general disadvantage of supervised ML approaches (Ruske et al., 2018). Random forest models, on the other hand, i.e. ensembles of multiple decision trees, allow for the over fitting aspect to be relaxed and adhere a higher level of generalization, due to the





*random* statistical methods used for model construction. Contrary to basic decision trees, only a *random subset* of predictors is used at each decision node within a tree. Besides, only a *random subset* (bootstrap sample) of the entire training data set is used to grow the decision tree during training, through sampling with replacement, allowing individual particles to be selected multiple times or not at all. Thus, randomness is introduced through both choosing the predictors used for decision splits and

bagging the training data. In fact, each tree of the random forest ensemble constitutes an independently trained model, grown on an equally sized, independent sample from the entire training data set. The unsampled particles of the training data set are referred to as out-of-bag (OOB) observations. These OOB provide a means to ensure training and testing is not performed on the same particle data, allowing for model cross-validation, i.e. estimation of the classification error of the model. This is achieved through prediction of particle class for these unsampled data and comparing to the true particle label (see SI Sect.

S7). In our model, each particle is ultimately assigned a class from each tree within the random forest and final particle type prediction is derived from the most frequently chosen, equally weighting the predictions of all trees.

For building the random forest model for the PPD-HS data, the calibration data sets of all target classes (particle shapes) were processed to calculate the PbP matrix of *parameter* values. Particle usability was checked using a minimum variance criterion of $2.8$ (see Appendix B). Particles not fulfilling this criterion were rejected from the respective target class and assigned as

noise. In addition, particles were required to have a minimum mean intensity of $3$ and a positive PTM value along either array, in order to discard particles affected by bad CMOS backgrounds (see SI Sect. S5.1).

In order to identify those parameters which are best suited for particle shape determination out of the pool of $n$ parameters calculated from the scattering pattern data, describing the data cloud in an $n$ dimensional parameter space, we have performed a principal component analysis (PCA, see Appendix C). PCA results revealed that $TBC_1$, $TBC_2$, $\Delta TBC$ and AIC yield

the most robust shape analysis. We therefore trained our random forest model using a matrix of feature vectors constrained to $TBC_1$, $TBC_2$, $\Delta TBC$ and $AIC$, which we refer to as shape indicators in the following. Using these four predictors, we trained a random forest model on $400,000$ randomly selected particles (training part), constituting of equal fractions of spherical and aspherical particles and using a total number of $200$ trees (see SI Sect. S7). Classification performance was then tested on the remaining particle data (test data set; $4,371,162$ particles), and subsequently applied to simulated hydrometeors

from our HINC experiments.

## 4  Results and Discussion

### 4.1  Particle sizing

As stated above, particle size is inferred from the signal recorded by the photodiode, by relating the signal intensity to the scattering cross section of the particle, using Mie theory, assuming spherical particle geometry and an isotropic refractive

index. For spherical particles the diameter value can usually be calculated with reasonable accuracy. Aspherical particles, however, are ascribed a spherical equivalent size based on the photodiode intensity value and thus can be mis-sized (Borrmann et al., 2000). In Fig. 4 we show a comparison of the optical particle size obtained by PPD-HS and the corresponding mode aerodynamic diameter from the APS, for the calibration data sets obtained with the setup shown in Fig. 3 for those within





the APS size range ($0 - 20$ μm, see SI Tab. S1 for details). We find good agreement of both optical and aerodynamic size for spherical and aspherical particles. It should be noted that even for the aspherical $NaCl$ particles, where the relationship between optical and aerodynamic size is complicated by an (unknown) shape factor, we find reasonable agreement between the sizing of PPD-HS and the APS. From these measurements we conclude that PPD-HS correctly sizes particles in the APS size range

5 up to approximately $20$ μm.

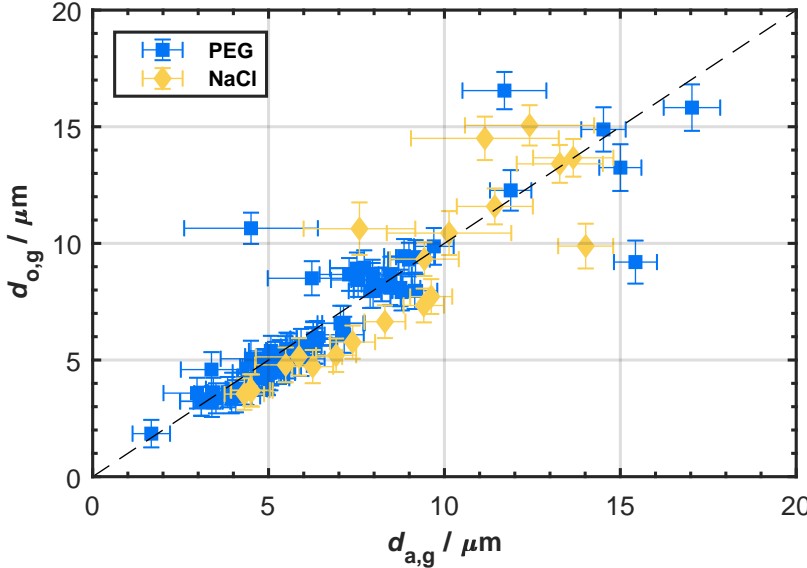

**Figure 4.** Particle sizing of PPD-HS and APS for spherical (PEG) and aspherical (NaCl) particles, showing the geometric mean of the optical diameter determined by PPD-HS as a function of the geometric mean of the aerodynamic diameter obtained from the APS, using the calibration setup shown in Fig. 3. Vertical and horizontal error bars indicate the geometric standard deviation of the optical and aerodynamic mean size, respectively. Data points outside the size range of the APS are not shown. A complete list of the individual data sets, including those covering sizes $> 20$ μm, is given in SI Tab. S1.

This is further supported by comparing the particle sizes determined by PPD-HS, which we refer to as instrument response (AD), to theoretically predicted sizes for the PEG particles using Mie theory and taking into account the optical geometry of PPD-HS (see SI Sect. S8). In Fig. 5 we depict the final calibration curves for the particle types used here, showing instrument response as a function of particle diameter.

10 **4.2 Particle shape classification: random forest model**

In Fig. 6 we provide the classification results, when the trained random forest model is applied to the test data. The confusion matrix is derived from comparing the prediction of the model against the true particle type. In this matrix, the diagonal cells (green boxes) show the number of particles that are correctly classified and indicate the corresponding percentage from total number of particles in the test data set. The dark gray cell at the bottom right, indicates the overall model *accuracy*, defined





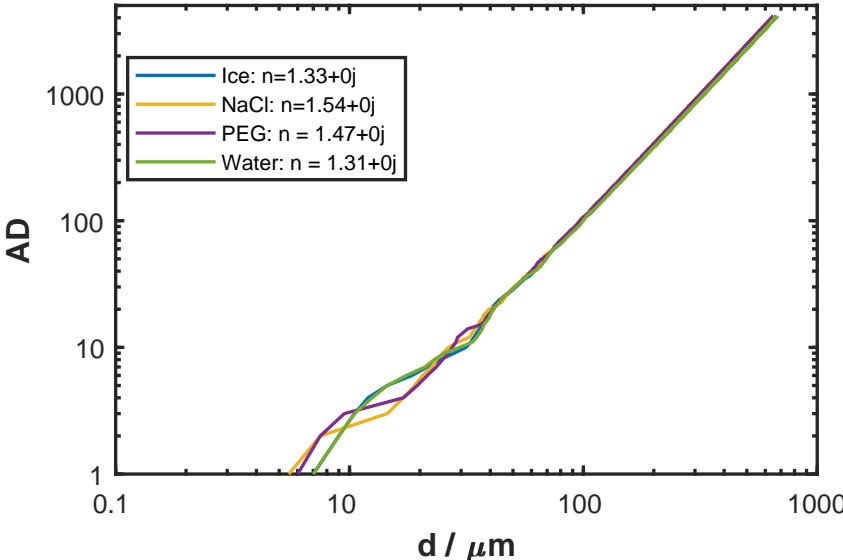

**Figure 5.** PPD-HS instrument response (AD) as a function of particle diameter for different particle types measured in this study. AD is a function of particle scattering cross section and instrument properties such as photodiode sensitive, signal amplification and laser power (see SI Sect. S8).

as the ratio of correctly predicted particles to the overall number of particles classified by the model. We find a high overall model accuracy with a true positive rate of 95.6 %, i.e. a good discrimination of particle shape when applying our random forest model to test data set.

However, in case of an imbalanced number of particles making up the individual classes of the test data set, the overall
model accuracy yields a biased picture, as the class with the largest number of members will dominate the counting statistics. Since our calibration data set encompasses a larger number of spherical particles compared to the aspherical particles, it is more meaningful to assess the model performance for each class separately, yielding a per-class accuracy. These per-class accuracies are indicated in the light gray, bottom row cells and indicated by the number of correctly predicted particles over the true number of particles within a target class. For instance, $592, 168$ particles are correctly classified as being aspherical,
resulting in 80.3 % of all particles belonging to the (true) aspherical target class to be correctly classified by the model. Higher classification performance is achieved for spherical particles with a per-class accuracy of 98.7 %. We interpret the lower per-class accuracy of the aspherical class to result from small scale aspherical features of some particles, which may result in calculated parameter values (TBC, $\Delta TBC$, AIC) comparable to those of spherical particles. Yet, another way to quantify model prediction power is achieved by evaluation of the model precision. Model precision is given by the values on the right-
hand side, light-gray column, indicating the percentages of all particles predicted to belong to a class, that are correctly (green) and incorrectly (red) predicted. We will refer to the incorrectly predicted values as false discovery rate (FDR). For instance, the FDR of the aspherical particle class is 7.2 %. Hence, out of all particles predicted to belong to the aspherical class, 7.2 % are





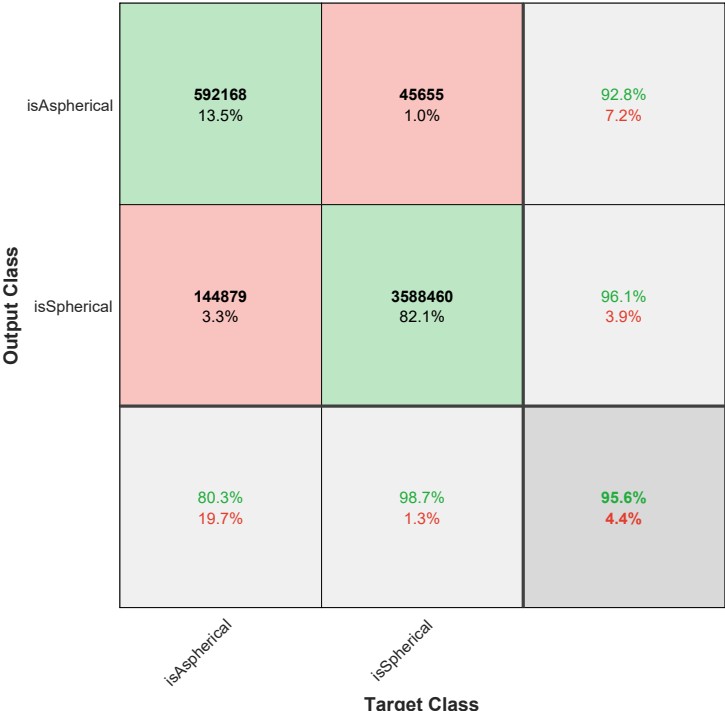

**Figure 6.** Confusion matrix of the random forest model applied to test data, i.e. the fraction of particles not used for model training. The random forest model was trained using 4 predictor variables and 200 trees. The confusion matrix shows the true (target) class (columns) of the particles versus the predicted (output) class (rows). **Coloured cells:** In each cell, the number of particles and percentage from total is given. **Precision:** The right-hand side, light gray column indicates percentages of all the particles predicted to belong to each class that are correctly (green, true discovery) and incorrectly (red, false discovery) predicted. **Per-class accuracy**: Light gray, bottom row cells give the percentages of all particles belonging to each class that are correctly (green, true positives) and incorrectly (red, false negatives) classified. **Overall accuracy:** Values shown in the bottom right cell give overall model accuracy, i.e. the fraction of particles correctly (green) and incorrectly (red) predicted out of all particles classified by the model.

incorrectly classified. Since the model precision is a measure of the closeness of repeated classifications by the model and does not involve a direct comparison to the true particle label, usage of the FDR is meaningful, as it directly yields an uncertainty for our model predictions, when used to classify particles of unknown label. Assuming a data set independent FDR, the number of wrongly predicted particles for each class can be calculated from the total number of particles predicted to belong to that class.

5 ## 4.3 Coupled HINC-PPD-HS measurements

To further test the performance of PPD-HS, we coupled it to HINC for detection of simulated cloud hydrometeors. Changing the thermodynamic conditions within HINC allows simulation of clouds containing only ice crystals, only supercooled liquid cloud droplets or a mix of the two, akin to MPCs, depending on the aerosol, $T$ and $RH$ used in HINC. In the following, the





results of three experiments using $NH_4NO_3$ and illite NX as seed aerosol are presented by applying the validated classification method derived from the calibration particles.

In Fig. 7 we show the results of an experiment at $T = 223$ K, using $NH_4NO_3$ aerosol. In Fig. 7a the temporal evolution of $T$ and $RH$ along the center of HINC (where the aerosols are injected) is shown, representing a typical $RH$ scan within a CFDC.
It should be highlighted again, that cloud particles in HINC are nucleated on the injected seed aerosol, but that supercooled liquid cloud droplets can only form once conditions of $RH_w \geq 100$ % are reached within the chamber at $T > 235$ K. Thus the measurements in Fig. 7 illustrate the response of PPD-HS to a pure ice cloud, established through homogeneous freezing of solution droplets formed by the $NH_4NO_3$ particles. The experiment starts at low $RH_w$ when the inlet valve is opened and $NH_4NO_3$ particles are introduced into HINC. As the $RH$ is increased within the chamber the $NH_4NO_3$ particles grow
hygroscopically and form solution droplets. Ice crystals ultimately start to form, above homogeneous freezing conditions, as indicated by the red, dashed line in Fig. 7a, where the grey enveloping shading indicates the uncertainty in $RH_w$ across the aerosol layer in HINC (Mahrt et al., 2018). At $\approx RH_w = 97$ % (11:47), where the particles detected by PPD-HS sharply increase, as displayed in panel b, the first ice crystals that formed via homogeneous freezing have grown large enough to be detected. The delay of observed homogeneous freezing in our experiment compared to the theoretical predictions from the
water-activity-based homogeneous freezing parameterization of solution droplets by Koop et al. (2000) can likely be explained by particles initially being below the detectable size of PPD-HS. In Fig. 7b we further show the particle type classification, as determined when applying the random forest model to the ice nucleation data. In the early stages of the experiment, particle classification is noisy revealing strong fluctuation of the individual particle type fractions. After 11:49 the majority of the particles is correctly classified as ice particles. We have highlighted two periods from this ice cloud experiment, as indicated by the
vertical lines in Fig. 7, representing distinct periods at the beginning and end of the experiment, where particle misclassification is high and low, respectively. In Figs. 8 and 9 we show the corresponding scattering patterns of a random collection of particles in chronological order. Form the scattering patterns shown in Fig. 8 it becomes clear that many ice particles show symmetric scattering patterns, i.e. small, freshly nucleated ice crystals appearing optically spherical. In case the c-axis of a hexagonal ice crystal is perfectly aligned with the optical axis of the image laser and at the same time the ice crystal is oriented in such a way,
that a symmetrical diffraction pattern is impinged on the detector arrays, low TBC and AIC values can result, which would cause the corresponding particles to get classified as spherical by our random forest model (see SI Fig. S1). Nevertheless, while this can cause some ice crystals to produce symmetrical scattering patterns, it is unlikely to explain all optically spherical ice crystals at low $RH$ conditions in our experiments.

The optical properties of ice particles have been reported to depend on the formation process, ice crystals nucleated from the
vapor phase generally depicting a higher degree of (optical) asphericity, compared to liquid-origin ice crystals, where smooth frozen droplets can form (Järvinen et al., 2016). While this would be in line with our homogeneous freezing experiment, where the solution droplets initially freeze into spherical frozen water droplets, we cannot exclude the formation of droxtals, i.e. frozen water droplets with faceted surfaces, thus complex crystals whose asphericity cannot be resolved by our instrument.

Specifically, similarly high spherical fractions were found at low $RH$, when forming pure ice clouds heterogeneously on
illite NX dust particles (not shown), i.e. ice crystals nucleated from and grown in the vapor phase. Thus, we attribute the





observed misclassification to the small optical particle size at these low $RH$ values, at which particle asphericity cannot be resolved by our instrument, consistent with the findings of Korolev and Isaac (2003), who report ice particle roundness to be mainly a function of particle size. In fact spherical ice particles, commonly reported for cirrus clouds (e.g. Garrett et al., 2005), can result from insufficient resolution of the optical probes deployed (Nichman et al., 2017). Finally, potential sublimation of

the small scale complexity (e.g. small facets) associated with the ice crystals after exiting HINC and before being imaged by PPD-HS can decrease the asphericity of the particles (Järvinen et al., 2016).

   Nevertheless, we note that there are particles classified as isAspherical, even though the scattering pattern appears symmetrical. This likely results from using the reduced information (shape indicators) for particle classification within our random forest model. For instance, high TBC values are not completely exclusive for aspherical particles as can be seen from the overlap-

ping probability distribution of the TBC predictor shown in SI Fig. S13. This results likely from spherical particles where the symmetric scattering patterns show a (slight) offset to the midpoint pixel (see SI Fig. S15). Conversely, we also observed some NaCl particles to appear symmetrical (low TBC value) on one of the detector arrays (see SI Fig. S16), but usually not on both (see SI Fig. S14). Thus using the information from two independent arrays should largely avoid and/or reduce misclassification of such particles. Overall, we cannot completely exclude the presence of artifacts within our training data sets, for instance

NaCl particles producing symmetrical scattering patterns, which are not removed by our usability criteria. This would require manual inspection of every particle within the calibration data set. This approach is time consuming and impractical for the number of particles sampled and not done here, given the overall good classification of spherical and aspherical particles with the VOAG.

   At approximately 11:51 the aspherical fraction starts to dominate, consistent with the ice particles having established suffi-

cient asphericity, during diffusional growth as the $RH_\mathrm{w}$ is increased within HINC and non-spherical features emerge. Using the corresponding size distribution from the RT-electronics for the time period around 11:51 (Fig. 7e), when the majority of the particles get correctly classified as aspherical, we find a minimum optical particle size of approximately 3.2 µm to detect asphericity (see SI Fig. S19). After 12:00, negligible observations are made of maximum TBC values below 0.05 (Fig. 7c; see also SI Fig. S13a-b), both the aspherical and spherical fractions stay almost constant at approximately 95 and 5 %, respectively.

The high aspherical fraction is supported by the asymmetric scattering patterns observed during period 2, depicted in Fig. 9.

   In Fig. 10 we show the results from a pure supercooled liquid sample formed at $T = 243$ K within HINC, using $NH_4NO_3$ aerosol particles. Water supersaturated conditions (panel a) are required to activate cloud droplets within HINC and grow them to sizes detectable by PPD-HS. As expected, no particles are detected by PPD-HS at $RH_\mathrm{w} < 100$ %, because these should be below the detection limit. From Fig. 10b it becomes immediately clear that the hydrometeors formed within HINC are

correctly classified as spherical particles, consistent with the absence of freezing of supercooled liquid cloud droplets formed on the $NH_4NO_3$ at this temperature. While the high spherical fraction could partly result from the optical particle sizes being constrained to approximately below 3 µm (Fig. 10e), the low maximum TBC values (Fig. 10c) along with the symmetrical scattering patterns (see SI Figs. S20 and S21) observed during the experiment are consistent with the classification results from our random forest model and at the same time reveal the power of using particle shape for phase analysis.





Finally, in Fig. 11 we depict results when using illite NX aerosol particles within HINC to simulate MPC conditions at $T =$ 238 K. The first ice crystals are heterogeneously nucleated on the dust particles around 17:45 at water subsaturated conditions, either through deposition nucleation or pore condensation and freezing (e.g. Kanji et al., 2017). Despite the large fluctuations at early stages of the experiment, the majority of particles is correctly classified as aspherical, with cloud droplets being absent

at $RH_w < 100$ %. The sharp increase in the number of detected particles after 18:10 coincides with water supersaturated conditions being reached and the co-existence of cloud droplets and ice crystals within HINC. The onset $RH_w$ of cloud droplet formation is consistent with that observed for the pure liquid cloud case discussed above (see Fig. 10). While the hydrometeor size distribution reveals diffusional particle growth as the $RH$ is increased (Fig. 11e), the distributions of AIC and maximum TBC values show a distinct bimodal character, neither observed for the pure ice nor the pure liquid cloud case (see Fig. 11c-d).

Here, the relatively large and small AIC values (and max. TBC values) are associated with ice crystals and cloud droplets, respectively, consistent with the mode of particles with AIC $> 0.05$ already appearing at water subsaturated conditions. On the contrary, the dominant mode at relatively small AIC values only evolves with the formation of cloud droplets, i.e. below $RH_w \approx 104$ % (18:13), usually required for the cloud droplets to grow to sizes large enough to become detected. At the same time it is interesting to note that this mode is capped at maximum TBC values of approximately 0.1 and with that comparable

to the values observed for the pure liquid cloud case (see Fig. 10c) and consistent with the distribution of TBC values from our calibration data set (see SI Fig. S13a-b). From this we conclude that both AIC and TBC are simple and powerful indicators that can be used to discriminate hydrometeor phase in MPC environments, as this bimodality is not as clearly observed in the particle size distribution (Fig. 11e), usually used for particle phase separation.





**Figure 7.** Freezing experiments of $NH_4NO_3$ aerosol using HINC at $T = 223$ K and a residence time $\tau = 22$ s. (a) $RH_w$ (blue line, left-hand ordinate) along with $T_{center}$ at HINC center line (orange line, right-hand ordinate), where aerosol particles are injected and hydrometeors are formed. (b) classification of particles detected by PPD-HS (left-hand ordinate) using random forest model (see Sect. 3.2) along with total number of detected particles (dashed, orange line, right-hand ordinate). (c) distribution of maximum TBC value and (d) AIC value of PPD-HS scattering patterns. (e) PPD-HS optical particle size distribution obtained from RT-electronics. Vertical solid lines indicate starting and end times of periods of interest, respectively.





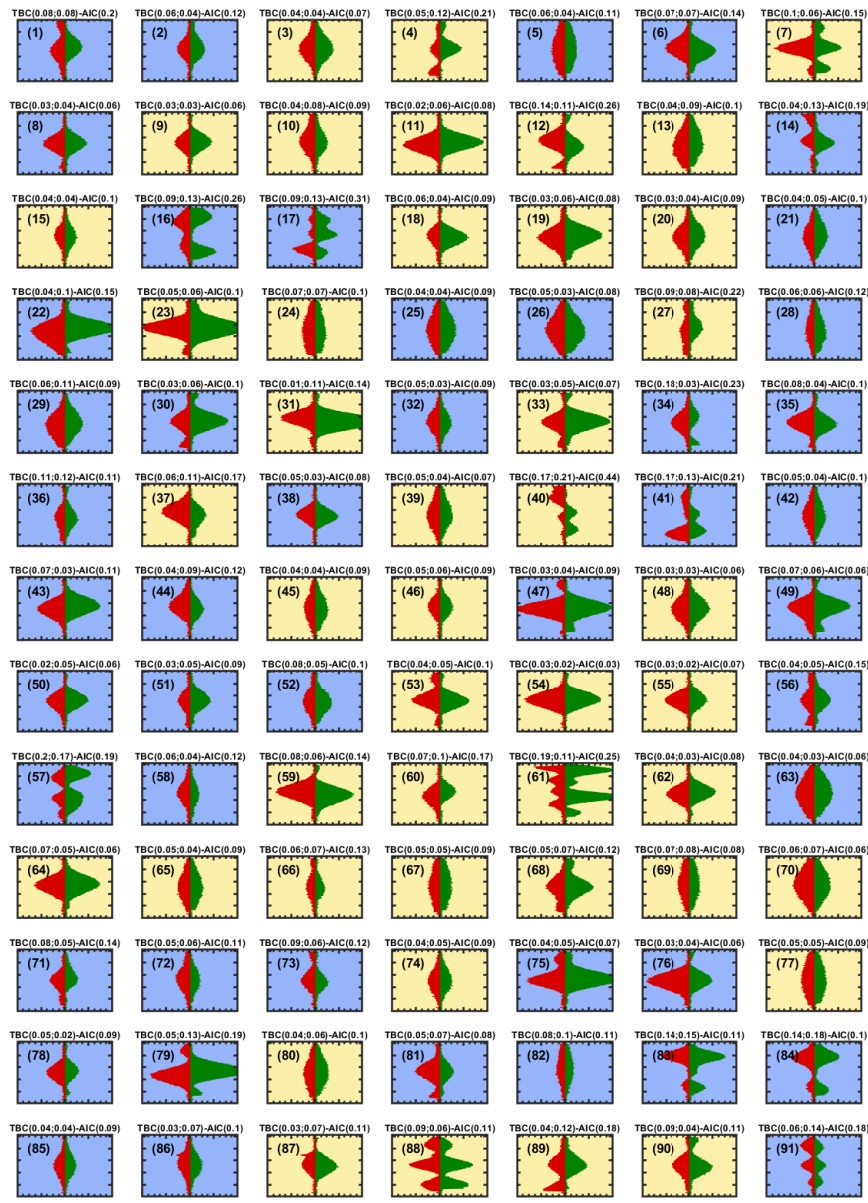

**Figure 8.** Collection of example particle intensity patterns imaged with PPD-HS during period 1 (where misclassification is high) of the experiment shown in Fig. 7. All scattering patterns are shown on the same intensity scale (40 a.u.). Background colour of the individual scattering patterns indicates particle classification by the random forest model into target classes isSpherical (blue) and isAspherical (yellow). The values on top of each panel depict the TBC, where the first number corresponds to array 1 and the second number to array 2, respectively, and the AIC.

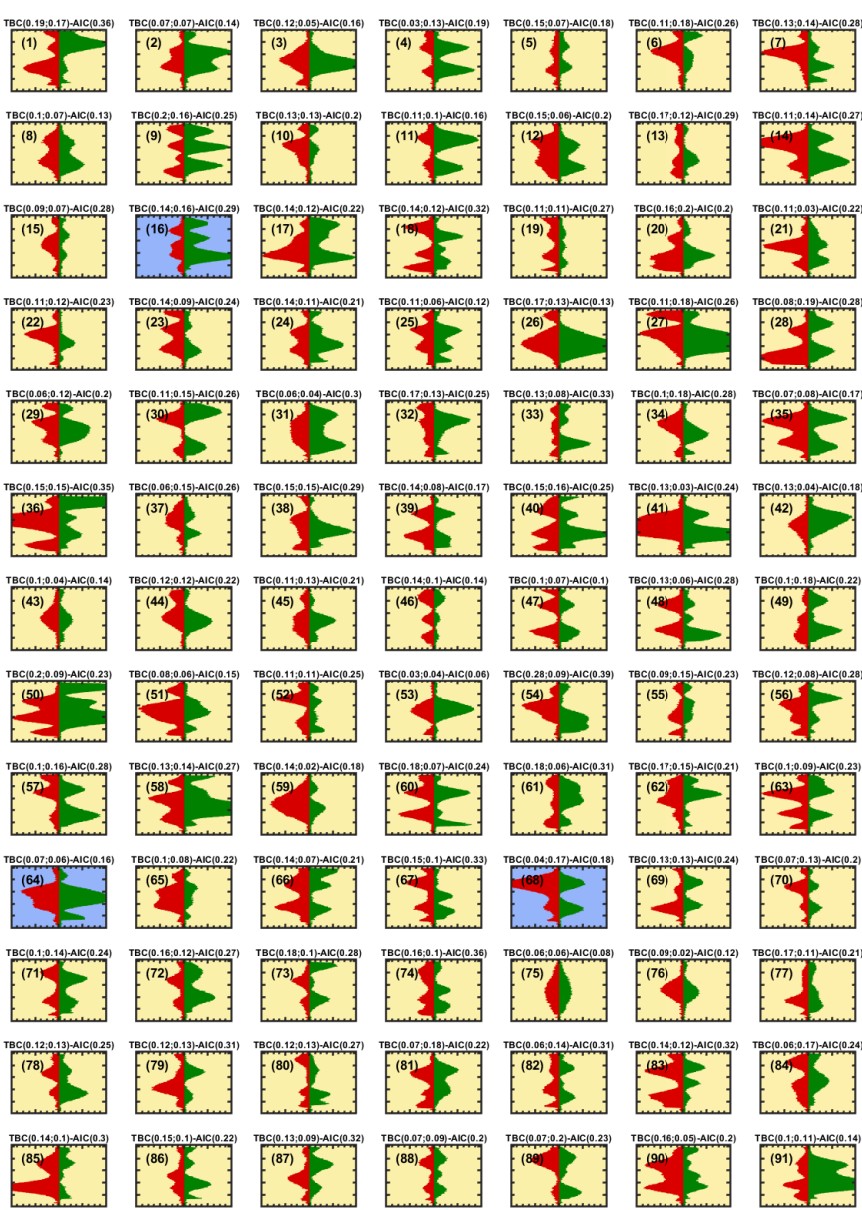

**Figure 9.** As in Fig. 8, but for particles imaged with PPD-HS during period 2 of the experiment shown in Fig. 7.





**Figure 10.** Freezing experiments of $NH_4NO_3$ aerosol using HINC at $T = 243$ K and a residence time $\tau = 22$ s. Panels and symbols as in Fig. 7.





**Figure 11.** Freezing experiments of $d_m = 400$ nm illite NX aerosol using HINC at $T = 238$ K and a residence time $\tau = 22$ s. Panels and symbols as in Fig. 7.





## 5 Limitations

In the current PPD-HS configuration the raw-electronics are only triggered after the RT-board is triggered by the photodiode. A direct triggering of both RT- and raw-electronics from the photodiode would allow for higher particle detection rates by the RT-electronics, desirable for laboratory experiments with usually high particle number concentrations and at the same time

5 allow the raw-electronics to contain (optical) particle size information. The latter could be used to more closely investigate the (a)sphericity of small ice crystals, as well as a size-dependent particle classification by the random forest model. In PPD-HS, the raw-electronics have the advantage of recording the complete scattering pattern, which allows the calculation of *any* particle parameters in a post-processing step, whereas the RT-electronics have the benefit that they could (theoretically) achieve higher particle detection rates than presented here, but at the same time are limited to the a priori specified parameters (such as TBC,

10 PTM etc.) that need to be calculated.

 Our coupled HINC-PPD-HS measurements are limited by the need for high supersaturations (or longer growth times) required to form ice crystals and cloud droplets of similar size coupled with the horizontal alignment of the setup. The fast growth kinetics for $T > 235$ K means the cloud droplets quickly grow by diffusion to diameters $> 3$ μm, where phase can (reliably) be determined by PPD-HS. However, for the residence times used in the experiments presented herein, these particles

15 are close to being lost by gravitational settling prior to reaching PPD-HS, resulting in an optimization between enough particle growth and loss due to gravitational settling before being sampled by PPD-HS. Such losses are circumvented by CFDCs with vertical orientation (e.g. Rogers, 1988; Stetzer et al., 2008; Garimella et al., 2016). However, water supersaturations would still be required to form MPCs and the $RH$ conditions would need to be optimized given the fixed residence time of such vertical setups. Vertically oriented chambers that allow for immersion freezing experiments (Lüönd et al., 2010; Kohn et al., 2016) or

20 atmospheric observations of hydrometeors, where cloud droplets and ice crystals of overlapping and large enough sizes can be formed, could both benefit from PPD-HS as a detector.

## 6 Conclusions

A major challenge in MPC analysis remains the discrimination between cloud droplets and ice crystals. Here, a new instrument, the High Speed Particle Phase Discriminator (PPD-HS), has been presented and characterized for sizing cloud particles and

25 determining their phase, with the goal to quantify the liquid and ice fraction in conditions relevant for MPCs.

 PPD-HS captures the near forward spatial intensity distribution of scattered light on a single particle basis. Different from previous devices, such as the PPD-2K, that use CCD cameras to capture the complete 2D scattering pattern, PPD-HS deploys two linear detector arrays, which capture a fraction of two 1D strips out of the complete scattering pattern. This reduction of the scattering data recorded and analyzed on a single particle basis, combined with the implementation of fast electronics used

30 to process this data allows for the high particle detection rates of several hundred particles per second. Symmetry analysis of these 1D scattering pattern is used to determine the shape of the light scatterer, which in turn is used to discriminate between spherical cloud droplets and aspherical ice crystals. Here, we introduced new shape indicators, the top to bottom comparison (TBC) and the array inter-comparison (AIC), that can be used to determine particle phase from symmetry analysis of the





scattering patterns captured by the two linear CMOS arrays. We presented a systematic instrument characterization of both particle size and phase determination in a well-controlled laboratory setup, which allows generation of nearly monodisperse spherical and aspherical particle populations, covering a size range of approximately $3-32$ µm using a vibrating orifice aerosol generator. Supervised machine learning was applied to the laboratory generated monodisperse calibration particles to

train a random forest model. Applying the trained model to a test data set of similar particles we demonstrated high overall classification accuracy, with the model correctly classifying 95.6 % of the particles. The classifier was subsequently used to classify simulated cloud hydrometeors sampled by PPD-HS in a set of CFDC experiments, using mineral dust (illite NX) and salt ($NH_4NO_3$) aerosol, where the phase of the hydrometeors can thermodynamically be predicted. The results discussed in this paper show that for the case of an ice crystal only sample flow, our random forest model incorrectly classifies the majority

of particles as droplets at early stages of $RH$ scan within the CFDC experiment, consistent with the symmetrical scattering patterns recorded during these experiments. We attribute this to small, optically spherical ice crystals formed within the CFDC. Thus, small ice crystals still remain a challenge for optical instruments. However, after $RH$ is increased and ice crystals have grown sufficiently, ice crystals are correctly recognized, yielding 3 µm as the lower size limit for the phase discrimination capabilities of PPD-HS. The misclassification rate is significantly lower in the case of a pure supercooled liquid cloud, where

spherical fractions of unity are predicted by our random forest model nearly throughout the experiment. This likely results from a less variable TBC distribution that is constrained to lower absolute values, but at the same time is likely biased by the limited droplet sizes ($< 3$ µm) achievable within our horizontal setup. To our knowledge, these data are the first of their type to be recorded on linear CMOS arrays, showing successful discrimination of spherical and aspherical cloud particles.

  To what extent PPD-HS can be used to determine the phase of atmospheric cloud particles remains to be investigated. It
is clear from this study that PPD-HS successfully discriminates between cloud droplets and ice crystals for particles $> 3$ µm when used with a CFDC setup, rendering PPD-HS an alternative to the size threshold criterion usually used with OPCs.

## 7 Code availability

The code version used for data post-processing and analysis in this paper is written in MATLAB and is available upon request to the authors.

## 8 Data availability

The data presented in this publication are available at the following DOI: https://doi.org/10.3929/ethz-b-000313787
Note by the authors: Data will be made available upon publication.





## Appendix A: Array midpoint determination

Determination of the midpoint $c_x$ of an array, i.e. center pixel, is crucial for correct calculation of shape indicators and thus ultimately phase discrimination. The midpoint for each array is determined empirically through calibration of PPD-HS with spherical particles of uniform size. For this purpose, all particles within a data set of spherical particles are considered as an
5   entity and the mean TBC of this entity is calculated for different midpoint pixels ($c_x$) using Eq. 2 and a range of $\pm 80$ pixels around the physical array midpoint ($p = 256$).

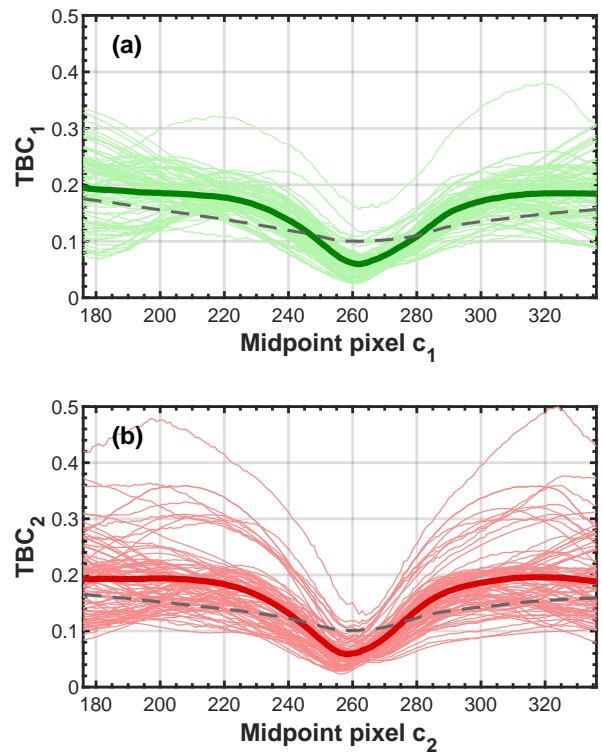

**Figure A1.** CMOS array midpoint determination from PPD-HS data of spherical particles. Mean TBC across all particles within a data set of uniformly sized particles for different CMOS pixels considered as midpoint within Eq. 2 for (a) CMOS array 1, (b) for CMOS array 2. All panels include the same data where light shaded lines correspond to individual data sets (see SI. Tab.S1) and the bold lines to the mean across all data sets. The gray, dashed line corresponds to the mean TBC across all data sets of aspherical particles for reference.

The midpoint pixel is then chosen to be the pixel yielding the lowest TBC value on average, since spherical particles producing symmetrical scattering patterns, should theoretically yield a TBC of zero. In Fig. A1 the mean TBC as a function of midpoint pixel is shown separately for both arrays. It can be seen that the mean TBC of the different spherical data sets
10  converges towards a minimum for a center pixel close to the physical array center. Using this method we determined the midpoint pixels to be 262 and 258 for array 1 and array 2, respectively, and all our data presented here is referenced to these



midpoints. For reference we include the mean TBC for our aspherical data sets (dashed gray line). It immediately becomes clear that for incorrect midpoint pixels (where TBC is not minimized for spherical particles), spherical and aspherical particles cannot be distinguished anymore using the TBC because the TBC of aspherical particles (gray dashed line) is smaller than that of the spherical particles (bold coloured lines) for incorrect midpoint pixels as seen in Fig. A1.

## 5  Appendix B:  PPD-HS data processing and analysis

Using MATLAB we developed routines to analyze data from PPD-HS. Each data set sampled by PPD-HS contains artifacts, which requires a careful usability check prior to phase discrimination analysis. In Fig. A2 we show an exemplary set of scattering patterns for a data set of spherical PEG particles of uniform size. While most particles show scattering patterns with clear features, some particles reveal very low or noisy, almost absent peak intensities. Scattering patterns with a low signal to noise ratio are caused by small particles that scatter minimally, or particles that miss the image laser beam completely due to an expansion of the particle sample flow or a miss match between trigger detection and pulsing of the image laser resulting from the (parabolic) velocity distribution of the particles. An indicator for the intensity features recorded by the two arrays is evaluated by the variance of the intensity along each array:

$$Var(I_\mathrm{x}) = \frac{1}{512} \sum_{p=1}^{512} (I_\mathrm{p,x} - \bar{I}_x)^2. \tag{B1}$$

Low variance values are mainly associated with noisy scattering patterns. Usable scattering patterns with a clear signal to noise on the contrary, are characterized through relatively higher variance, resulting from clear scattering features (distinct peaks) along each CMOS array. Comparing the variance distributions of data sets of spherical particles and mere electronic noise, detected by the CMOS arrays when no flow and thus no particles were present within the scattering chamber, we have empirically found a minimum variance value of 2.8 needed on each array, in order to be considered usable. A raw scattering pattern that does not fulfill this criterion is considered noise and is rejected from determining particle shape. Nevertheless, these noise patterns are included for concentration analysis, without classifying the particle type. For the scattering patterns shown in Fig. A2 we indicate for each particle, the values of TBC, AIC and variance of both arrays on top of each panel. Visual inspection reveals that particles 1,3, 9 and 11 are not suitable for phase discrimination, consistent with low variance values. Particle 13 shows weak scattering intensities, compared to other particles (e.g. 2, 10 and 12) but can still be associated with a spherical particle and has variance values above 2.8 on each array.

In Fig. A3 we show the effect of the scattering patterns usability check using a minimum variance criterion of 2.8 for a typical VOAG data set of spherical particles. TBC probability density functions (PDF) are constrained to lower TBC values when only particles with a minimum variance of 2.8 on either CMOS array are considered (dotted lines), compared to when all particles of the data set are used (solid lines). For instance, for the data set shown in Fig. A3, most of the particles with TBC $> 0.1$ are removed after applying the minimum variance criterion. Furthermore we note this example shows good agreement of the TBC distributions among the two CMOS arrays when using the respective midpoints discussed above.

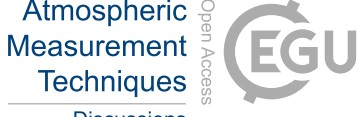



**Figure A2.** Example scattering patterns showing the relative scattered light intensity as a function of array pixel number, for array 1 (green) and array 2 (red). Data are background corrected and correspond to VOAG generated PEG particles of $d_{a,g} = 3.39$ µm. The values on top of each panel depict the TBC, AIC and variance, where the first number corresponds to array 1 and the second number to array 2.

In addition to this variance criterion other selection criteria can be applied to constrain the data, at the cost of reducing the size of the data set. This is particularly useful in cases where a sample is constituted of multiple particle types. For instance, the fly ash suspension contained residuals that resulted in scattering patterns that could be associated with spherical particles,





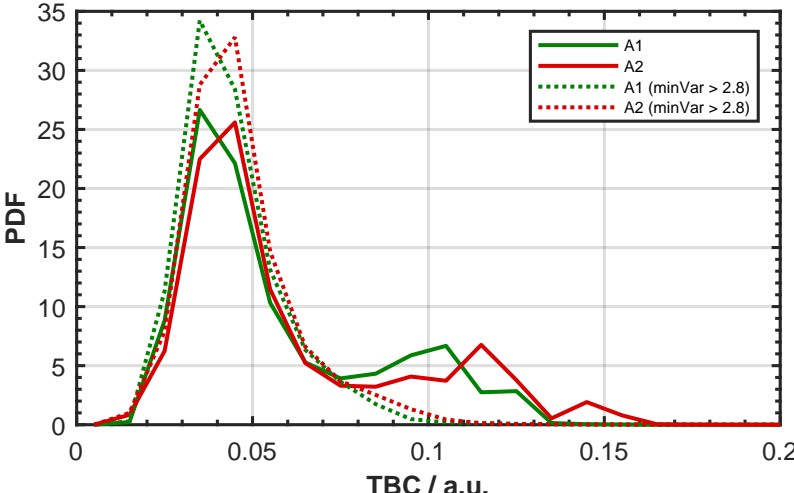

**Figure A3.** Example comparison of TBC distributions of a VOAG data set of spherical particles (5 μm PEG) when the minimum variance criterion of 2.8 is (dotted) and is not (solid) applied. Green lines correspond to array 1 and red to array 2.

such as displayed in Fig. 2a, and thus cannot be unambiguously attributed to fibres (see SI Fig. S2). Fibrous particles can heuristically be separated out from this data set by selecting only particles above a certain PTM threshold i.e. particles with a certain aspect ratio. Similarly, in order to avoid any bias in the calculated particle parameters from high CMOS background intensities (see SI Sect. S5.1), we categorized all particles as noise, which have intensities below $-2$ (see SI Sect. S5.2).

**Appendix C: Multivariate PbP data**

**C1    Principal component analysis**

The intensity data from the CMOS arrays can be used to calculate a user-defined number of parameters for each particle in a post-processing step, leading to a multivariate data set where each variable constitutes a dimension (degree of freedom) and each particle represents an observation. Here, we calculate a total number of 11 variables from the scattering pattern data. In
the phase-space described by these variables information is often correlated and thus redundant. Covariances of the variables describing the phase space of such a data cloud need to be considered when using ML techniques to classify particles, in order to obtain robust classification results.

Here, we use PCA on our multivariate data set. In general the purpose of PCA is detection of the dominant modes of variability, which are mutually orthogonal and uncorrelated. During PCA, redundant information in the form of variable correlation is
15 bundled by describing the PbP data using a set of new, linearly uncorrelated variables, so-called principal components (PC), which constitute linear combinations of the original variables, i.e. particle parameters (TBC, PTM etc.). Usually, a few PCs are sufficient to explain a large fraction of the total variance, so that describing the data in a subspace with fewer dimensions



(reduced to the dominant PCs) is adequate. Here, we are interested in reducing the dimensionality of our phase-space describing the PPD-HS data, prior to using supervised ML for particle shape classification, aiming at identifying variables which are suited to make statements about particle shape.

Mathematically, the PbP data are transformed into a new, orthonormal coordinate system described by the eigenvectors,
derived from an eigenvalue decomposition of the variance-covariance matrix, of the original PbP data matrix. That is, the eigenvectors are aligned along the symmetry axes of the data cloud, with the first eigenvector pointing into the direction of the largest data spread, the second eigenvector along the largest variability orthogonal to the first eigenvector and so on, with the total variance being preserved upon coordinate transformation. In order to derive meaningful results, it is important to normalize the original PbP data matrix prior to performing PCA. For instance, for the size range covered here, the mean light intensity shows a larger variability compared to the TBC, since the latter one is normalized to the maximum light intensity and with that independent of physical particle size, whereas the mean light intensity is not. Thus, without normalization, the variability in the mean light intensity would lead to a stronger weighting of this variable upon PCA, compared to variables with less variability (such as e.g. TBC). Therefore, a z-score standardisation on the variables is performed here prior to PCA, subtracting the average from each data value and dividing by the standard deviation, so that a similar emphasis is given in all
phase-space directions upon PCA.

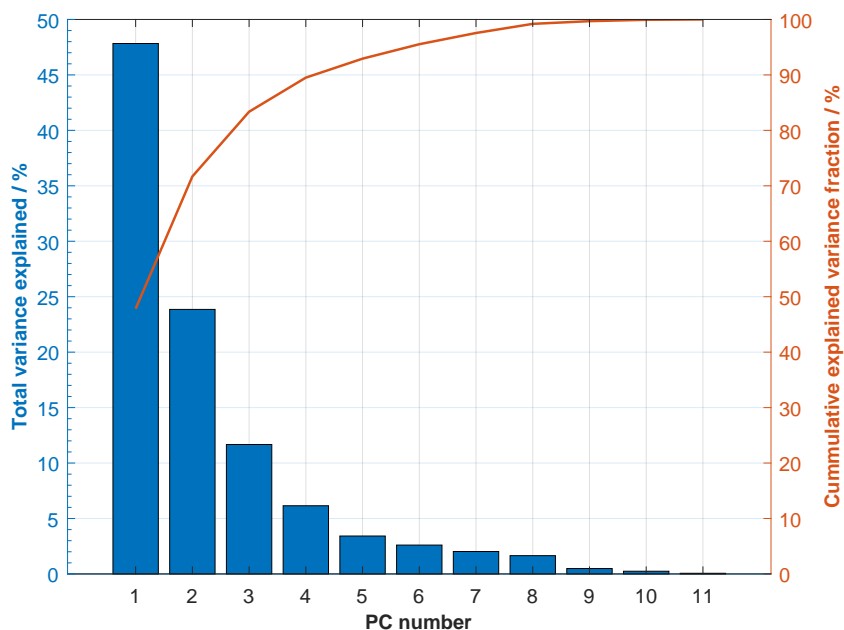

**Figure A4.** Fraction of variance explained by each PC (left hand ordinate), listed in Tab. A1, and cumulative explained variance (right hand ordinate). PCA was performed on a subset of $10,000$ particles, randomly selected from the entire calibration data set and consisting of equal numbers of spherical and aspherical particles.





In Fig. A4 we show the variance explained by each PC, obtained when performing a PCA on the normalized PbP data matrix, using a total of $10,000$ particles that were randomly selected from the entire calibration data set, but encompassing an equal number of spherical and aspherical particles (target classes). The first four PCs, listed in Tab. A1, describe over $90$ % of the total variance, thus are associated with high eigenvalues.

| Variable | PC 1 | PC 2 | PC 3 | PC 4 | PC 5 | PC 6 | PC 7 | PC 8 | PC 9 | PC 10 | PC 11 |
|---|---|---|---|---|---|---|---|---|---|---|---|
| $I_{max,1}$ | 0.41 | 0.02 | 0.09 | 0.14 | -0.32 | 0.02 | 0.05 | -0.41 | -0.33 | 0.61 | 0.23 |
| $I_{max,2}$ | 0.39 | 0.04 | 0.12 | -0.28 | -0.44 | 0.02 | -0.14 | 0.29 | -0.47 | -0.42 | -0.24 |
| $\bar{I}_1$ | 0.43 | -0.03 | -0.03 | 0.04 | -0.10 | -0.02 | 0.04 | -0.22 | 0.56 | 0.06 | -0.66 |
| $\bar{I}_2$ | 0.43 | -0.02 | -0.04 | -0.06 | -0.17 | 0.00 | -0.05 | 0.15 | 0.52 | -0.23 | 0.66 |
| $TBC_1$ | 0.01 | 0.53 | -0.24 | -0.17 | 0.07 | 0.77 | -0.16 | -0.11 | 0.02 | 0.02 | 0.00 |
| $TBC_2$ | 0.00 | 0.54 | -0.22 | 0.21 | 0.00 | -0.51 | -0.59 | -0.07 | -0.01 | -0.02 | 0.00 |
| $PTM_1$ | -0.02 | 0.26 | 0.62 | 0.67 | -0.07 | 0.21 | 0.05 | 0.18 | 0.06 | -0.12 | -0.03 |
| $PTM_2$ | 0.00 | 0.24 | 0.67 | -0.58 | 0.25 | -0.16 | -0.09 | -0.18 | 0.11 | 0.10 | 0.04 |
| $VarI_1$ | 0.39 | -0.02 | -0.04 | 0.18 | 0.57 | -0.02 | 0.14 | -0.43 | -0.26 | -0.46 | 0.06 |
| $VarI_2$ | 0.39 | -0.02 | -0.04 | 0.02 | 0.51 | 0.02 | -0.10 | 0.63 | -0.08 | 0.41 | -0.06 |
| $AIC$ | 0.03 | 0.55 | -0.17 | -0.08 | -0.06 | -0.27 | 0.75 | 0.16 | -0.01 | 0.01 | 0.00 |

**Table A1.** PC coefficients, also known as loadings or eigenvectors, derived from performing a PCA on the standardized (mean centered and normalized to standard deviation) PbP data matrix of original particle variables. PCA was performed on $10,000$ particles composed of an equal number of aspherical and spherical particles, randomly selected from the entire calibration data set. Each column contains the coefficients for one PC, which are given by a linear combination of the original variables (rows).

## 5 C2  Identification of robust particle shape indicators

Using the PC coefficients depicted in Tab. A1, we transformed the PbP data matrix of the randomly selected $10,000$ particles containing the original particle variables (TBC, PTM, AIC ..., see first column in Tab. A1) into the phase space described by the eigenvectors of the PCA. The transformed data matrix was subsequently used to train a random forest model, as described in Sect. 3.2. This was done ten times, each time randomly sampling $10,000$ particles from the entire calibration data set (see SI

10 Tab. S1), but using the same PC coefficients (see Tab. A1), in order to test the robustness of the model and estimate predictor importance, with the goal to identify robust particle shape predictors.

Figure A5 shows boxplots of the predictor importance for particle shape discrimination, derived from the PC based random forest model and estimated using a curvature test. In the curvature test, the best predictor is determined through minimization of the $p$-value for evaluation of the null hypothesis that predictor and response are independent, as detailed in Loh and Shih

15 (1997), at every decision node within a tree. The predictor importance can thus be viewed as a measure for how well a given predictor splits the data, out of the randomly selected set of predictors considered at a decision node. The second and sixth





PCs show a significantly lager predictor importance compared to the other PCs, thus best discriminate particle shape out of the entire set of predictors (PCs) that is used as input for our random forest model.

This observation is consistent with the PC coefficients depicted in Tab. A1. We interpret PC 2 as shape component (variability), consistent with the relative strong contribution of coefficients, that describe the symmetry of a scattering pattern, namely $TBC_1$, $TBC_2$ and AIC. In contrast, in PC 6, the signs for the TBC coefficients show an opposing trend. We interpret this PC to describe particle shape in terms of a $\Delta TBC$, given by the difference between the TBC values obtained from the two CMOS arrays. This value is small for spherical particles with similar scattering patterns across both arrays and larger for aspherical particles, where the intensity distribution is not symmetric.

From the PCA we conclude that the symmetry parameters $TBC_1$, $TBC_2$, $\Delta TBC$ and AIC are best suited to discriminate the shape of particles detected by PPD-HS and that the PbP input for our classification model should be constrained to these predictors. We note the contributions of the PTM coefficients in PC 2. However, we cannot think of any physical contribution of the PTM to distinguish between spherical and aspherical particles and thus do not use this parameter within our random forest model.

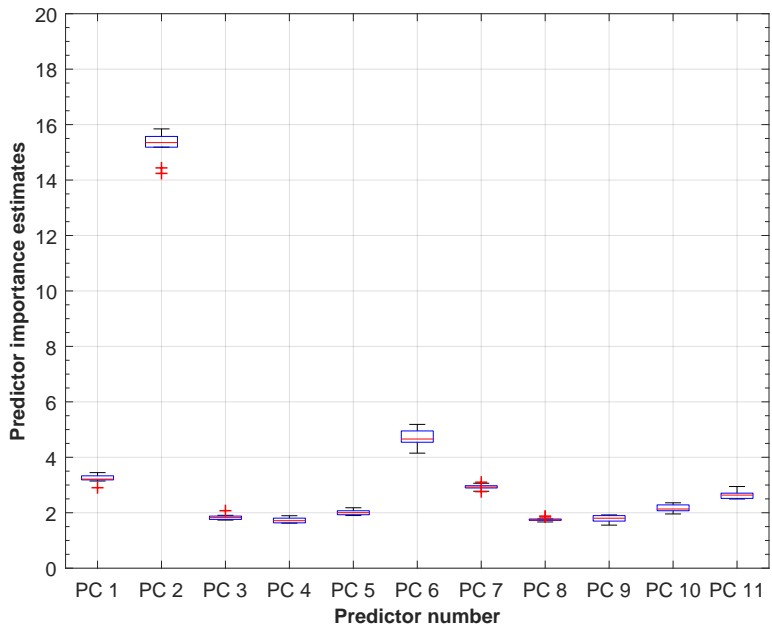

**Figure A5.** Estimates of predictor importance derived from random forest model using the PCs as model input (see Tab. A1) encompassing $10,000$ observations composed of equal numbers of spherical and aspherical particles, randomly pulled from the entire calibration data set. The random forest model was grown on 200 independent decision trees and predictor importance was derived from a curvature test for predictor splitting (Loh and Shih, 1997) using the OOB samples. Each box encompasses a total of ten independent simulations (random forest models), with the red line representing the median and the bottom and top edges of the box indicate the $25^{th}$ and $75^{th}$ percentiles, respectively. Whiskers extend to the most extreme data points that are not considered as outliers, which are given by the red crosses.



*Author contributions.* FM prepared all figures and wrote the manuscript with contributions from all authors. The schematic illustrating the experimental set up was created by JW. PPD-HS was built by CS and purchased by ETH, where detailed characterization measurements were performed. FM designed the calibration measurements and FM and ZAK designed the ice nucleation experiments. JW and FM conducted PPD-HS experiments with some help of RD. JW, RD and FM analyzed PPD-HS measurements and developed data analysis routines. HS performed theoretical Mie calculations. RD, JW, FM, CS, HS and ZAK discussed and interpreted data. ZAK supervised the project.

*Competing interests.* The authors declare that they have no conflict of interest.

*Acknowledgements.* Fabian Mahrt and Zamin A. Kanji acknowledge funding from ETH grant application ETH-25-15-1. Zamin A. Kanji acknowledges funding from the ETH Scientific Equipment Program.We would like to thank Sarah Grawe for providing the fly ash sample and help with the production of fibre-like particles. Hannes Wydler, Peter Isler and Marco Vechellio are acknowledged for technical support throughout the project. Fabian Mahrt acknowledges Carsten Kykal for advice about using the VOAG. The authors thank Ulrike Lohmann, Caroline Rösch and Zbigniew Ulanowski for reading the manuscript and for helpful discussions. Niklas Pfister is thanked for helpful input and discussions on the supervised machine learning approach.





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
