# Peer review of "A High Speed Particle Phase Discriminator (PPD-HS) for the classification of airborne particles, as tested in a continuous flow diffusion chamber"

_Atmospheric Measurement Techniques, 2019_

## Referee Comment (RC1) · Anonymous Referee #1 · 28 Feb 2019

**Reviewer's comments**

**A High Speed Particle Phase Discriminator (PPD-HS) for the classification of airborne particles, as tested in a continuous flow diffusion chamber** by Mahrt et al. [**AMT-2019-36**]

Page and line numbers refer to submitted manuscript (22 February 2019)

**General comments:**

The authors present a new method to characterise the particle phase of hydrometeors using a novel light scattering instrument and machine learning algorithm. This represents a promising advance in studying mixed phase cloud microphysics since particle phase can be determined from scattering parameters independent of the particle size. The utility of the method is demonstrated using different conditions in a CFDC to generate liquid droplets, ice crystals, and mixtures.

The manuscript is very well written and scientifically sound. I recommend that it is suitable for publication in its present form, however I have a few minor suggestions for the authors to consider, listed below. I think there is great potential for more detailed investigations of cloud microphysics with this technique, and look forward to (hopefully) future measurements of mixed phase clouds in the atmosphere. I would also like to commend the authors on the excellent documentation, data, and explanations of their methodologies and measurements provided.

**Specific comments:**

P5 Fig1: I would suggest clarifying that the light orange and brown shading denote *detected* light from the trigger and image laser beam.

P6 L9: Can you provide an estimate of the azimuthal angle range of detected light?

P6 L6-9: What is the purpose of L4-L5?

P6 L11-12: Was saturation of the CMOS arrays ever problematic

P9 L1-4: Since $\Delta TBC$ is later determined to be a key value, perhaps it is worth mentioning it here.

P11 L3-4: Where/how were temperature and relative humidity measured?

P14 Fig5: I believe the minimum value on the x-axis should be 1μm instead of 0.1 μm.

P16 L22-23: Can you comment on why the machine learning algorithm would classify particles as aspherical (e.g. Fig8 39,46,74) with lower TBC and AIC values than particles classified as spherical (e.g. Fig8 29, 57, 41)?

P17 L34: Can you offer an explanation for the trimodal appearance of the AIC values for the 243K $NH_4NO_3$ case (Fig 10d)?

SI P4 FigS4: It would be helpful to have *RTe* and *RAWe* defined in the caption (presumably real-time electronics board and raw sampling electronics?).

SI P6 FigS6: "*2 μm*" instead of "*2 mum*"

SI TableS1: A minor suggestion, it would be helpful to have the data sets used in FigS12 highlighted by bold or coloured font.

---

## Referee Comment (RC2) · Anonymous Referee #2 · 2 Apr 2019

**A High Speed Particle Phase Discriminator (PPD-HS) for the classification of airborne particles, as tested in a continuous flow diffusion chamber** by Mahrt et al.

The authors present a new instrument, a High Speed Particle Phase Discriminator (PPD-HS), for phase discrimination of cloud particles. The instrument is an extension to the Small Ice Detector (SID) family but compared to the SID-3 and the Particle Phase Discriminator (PPD) it uses two CMOS arrays instead of a CCD camera. This modification allows recording of two 1D rows of scattering information, which reduces the amount of recorded data and allows higher detection rates of several hundred particle per second. The 1D scattering patterns are analyzed for their symmetry to discriminate between spherical and aspherical particles. The authors present characterization of the new instrument and have developed a supervise machine learning algorithm to automatically classify particles for their phase. The paper is well written and the used approach justified. However, few aspects should be addressed before publication.

**Major comments**

**1.** The random forest model was trained with a test particle dataset of droplets and NaCl particles. However, it is somewhat questionable if NaCl particles would be a good proxy for aspherical ice particles. Why the random forest model was not trained using real ice particles from the experiment where $NH_4NO_3$ aerosol was frozen at T = 223 K? Alternatively, would it have been possible to use SID-3 and PPD 3D scattering patterns of ice particles as training sets?

**2.** The authors do not discuss what is the upper size limit of the PPD-HS. Although the method is calibrated up to 32 $\mu$m using the test particles, the HINC experiments do not produce particles >10 $\mu$m. It is unclear how well the PPD-HS discriminates phase in the size range of 10-30 $\mu$m. For example, how would 1D patterns of large droplets with multiple rings be classified? Also, larger complex ice particles have 3D scattering patterns showing frequent speckles (as seen in Fig. 2b). How would the symmetry of such particles look like in the PPD-HS?

**3.** The dead time of the instrument (177 to 267 $\mu$s) is high and its consequences to sampling statistics are only discussed quantitively in the supplementary materials. However, this discussion should be incorporated into the main text. In chapter 5 the issue is mentioned but there is no discussion of the implications for future ice spectrometer and field measurements. If at a typical mixed-phase concentration of 100 cm$^{-3}$ 100% or more particles are missed, how representative are the retrieved PPD-HS ice concentrations, especially if looking for the first ice? Consequently, the last sentence in the abstract should be modified accordingly.

**Minor comments**

**p.2, line 18:** "This is despite the knowledge that cloud particle size distributions comprised of a mixture of cloud droplets and ice crystals are affected by the presence of small ice particles". Please provide references.

**p.2, line 34:** Spherical particles can change the incident polarization state. The incident polarization state remains unchanged in the case it is linearly polarized (horizontal or vertical). This is why depolarization techniques use linearly polarized light.

**p.3, line 1:** Depolarization techniques are sensitive to even small changes from a spherical shape, e.g. spheres with surface roughness or oblate/prolate particles do cause a measurable depolarization signal. However, commercial polarization sensors, like the one used in Zenker et al., 2017, can have difficulties discriminating small ice particles from droplets. These two techniques should not be mixed.

**p.3, line 3:** The ability to discriminate phase based on particle shape depends on the used method. Shadow imaging and optical microscopy in best case have an optical resolution around 2 $\mu$m and, thus, cannot be used to discriminate the phase of small (<50 $\mu$m) cloud particles. 2D diffraction patterns can reveal more details that cannot be resolved by traditional imaging methods.

**p.3, line 9:** small-scale complexity, please add the following references: Ulanowski et al., 2014; Schnaiter et al., 2016

**p.7, line 4:** The discussion of the electronic dead time is crucial for understanding the instrument performance and should be added to the main text. The dead time of 177 to 267 $\mu$s is high compared to other cloud instrumentation, which will lead to reduced sampling volume. Fig. S11 shows that at a typical mixed-phase concentration of 100 cm$^{-3}$ 100% or more particles are missed, which will have severe consequences for detecting ice in mixed-phase conditions.

**Fig. 2:** The example droplet (a) is a larger droplet with multiple visible rings that does not correspond with the scattering data from CMOS array (d) that shows 1 or possibly 2 maximums corresponding to a small droplet. How would scattering data from a larger droplet with multiple rings look like?

**p.11, line 20:** Are the calibration datasets from the VOAG experiments representative for cloud particles? First, the particle sizes are limited to 32 $\mu$m, whereas ice crystals can be significantly larger. More importantly, are salt particles a good proxy for ice crystal shapes? Why not use HINC experiments in cirrus conditions as training data sets?

**p.13, lines 4-5:** The scattering cross section of aspherical particles between 10.6° to 101.0° can be very different to NaCl. Therefore, it cannot be stated that PPD-HS correctly sizes all aspherical particles <20 $\mu$m.

**Fig. 4:** Please explain $d_{o,g}$ and $d_{a,g}$ in the figure caption.

**Fig. 7:** All the particle concentrations are given as counts and not as number concentrations within a volume. Since the sensitive area of the PPD-HS is known the instrument counts can be converted to concentrations. Also, it would be illustrative if the mean modal diameter would be marked in panel e.

**P. 24, line 8:** What would be the maximum data acquisition rate of the RT-electronics if using priori specified parameters?

**P. 25, line 12:** Please define small.

**References**

Schnaiter, Martin, et al. "Cloud chamber experiments on the origin of ice crystal complexity in cirrus clouds." *Atmospheric Chemistry and Physics* 16.8 (2016): 5091-5110.

Ulanowski, Zbigniew, et al. "Incidence of rough and irregular atmospheric ice particles from Small Ice Detector 3 measurements." *Atmospheric Chemistry and Physics* 14.3 (2014): 1649-1662

---

## Author Comment (AC1) · 23 May 2019

Reviewer comments are reproduced in **bold** and our responses in normal typeface; extracts from the originally submitted manuscript are presented in *red italic*, and from the revised manuscript in *blue italic*.

We have numbered the reviewer's major comments for ease of cross-reference within the other reviews.

**General comments:**
**The authors present a new method to characterise the particle phase of hydrometeors using a novel light scattering instrument and machine learning algorithm. This represents a promising advance in studying mixed phase cloud microphysics since particle phase can be determined from scattering parameters independent of the particle size. The utility of the method is demonstrated using different conditions in a CFDC to generate liquid droplets, ice crystals, and mixtures.**
**The manuscript is very well written and scientifically sound. I recommend that it is suitable for publication in its present form, however I have a few minor suggestions for the authors to consider, listed below. I think there is great potential for more detailed investigations of cloud microphysics with this technique, and look forward to (hopefully) future measurements of mixed phase clouds in the atmosphere. I would also like to commend the authors on the excellent documentation, data, and explanations of their methodologies and measurements provided.**

We thank the reviewer for carefully reading the manuscript and the overall constructive comments on it. We hope that the responses below satisfactorily address the reviewer concerns.

**Specific comments:**
**P5 Fig1: I would suggest clarifying that the light orange and brown shading denote *detected* light from the trigger and image laser beam.**
We have changed the last sentence in the caption of Fig. 1 to now read:
*"Light orange and brown shading in panel (a) and (b) correspond to light scattered by particles when passing the trigger laser beam and image laser beam, respectively, and ultimately detected by the PD and the CMOS arrays."*

**P6 L9: Can you provide an estimate of the azimuthal angle range of detected light?**
The azimuthal angle is 9°. We now add this information in the revised manuscript (P6L12):

**P6 L6-9: What is the purpose of L4-L5?**
We have added the following sentence on P6L9 (initial manuscript) P6L17 (revised manuscript) to explain the use of L4 and L5:
*"The lenses L4 and L5 reduce the size of the image independently in the horizontal and vertical planes, respectively, yielding the elliptical output image ultimately captured by the linear CMOS array system."*

**P6 L11-12: Was saturation of the CMOS arrays ever problematic.**
No, (intensity) saturation of the CMOS arrays has never been observed for the particle types and size range used in the experiments of the presented study.

From Fig. 5 in the revised version one can see that the maximum particle size that would trigger the laser is approximately 70 μm. At this size, the 12-bit detector reaches a maximum value, i.e. saturates (AD = 4096), as indicated by the instrument response (AD).

We have added the following statement on P13L9 (initial manuscript) P14L7 (revised manuscript):

*"It can further be seen that the a maximum AD is reached for particles of approximately 70 μm yielding an upper size limit for particles to be detected and recorded by PPD-HS (detector saturation). However, it should be noted that the maximum particle size tested here was 32 μm and that an upper size limit of PPD-HS would need to be tested in future experiments."*

**P9 L1-4: Since *ΔTBC* is later determined to be a key value, perhaps it is worth mentioning it here.**

We agree with the suggestion of the reviewer to mention the ΔTBC already at this point, or at least hint at the possibility of deriving further parameters from the two TBC values, which is revealed by the PCA analysis later in the manuscript. We have therefore added the following statement (page X line X in revised manuscript):

*"Moreover, the TBC values of both arrays can be used to derive further sphericity parameters, e.g. the ratio of both TBCs or their absolute difference, which can improve particle classification."*

**P11 L3-4: Where/how were temperature and relative humidity measured?**

The specified temperatures ($T$) and relative humidities ($RH$) denote the conditions within the horizontal ice nucleation chamber (HINC), which is used to produce either cloud droplets, ice crystals or a mixture of both on the injected aerosol particles. These are calculated values derived by controlling the temperature of ice-coated chamber walls. HINC is a continuous flow diffusion chamber and etailed descriptions of its working principle including the control of $T$ and $RH$ conditions can be found in the indicated references P10L18: Lacher et al. (2017), Mahrt et al. (2018). This method of exposing aerosol to defined $RH$ and $T$ conditions has been well established for a few decades (Hussain & Saunders, 1984; Rogers, 1988).

**P14 Fig5: I believe the minimum value on the x-axis should be 1μm instead of 0.1 μm.**

Thank you for spotting this. We agree, the limits of the x-axis are 0.1 and 100 μm and we have corrected the figure accordingly.

**P16 L22-23: Can you comment on why the machine learning algorithm would classify particles as aspherical (e.g. Fig8 39,46,74) with lower TBC and AIC values than particles classified as spherical (e.g. Fig8 29, 57, 41)?**

The reviewer raises a valid point here. When considering for instance the particles #29 and #39 of Fig. 8, both show a visually symmetric scattering pattern, and should hence be classified as spherical. The visual symmetry is described in terms of 4 absolute numbers, namely TBC1, TBC2, ΔTBC and AIC, which are used by the algorithm to determine particle shape. We agree that it appears counterintuitive that the particle with overall higher TBC and AIC values (#29) is classified as spherical, whereas the particle with the lower TBC and AIC values (#39) is classified as aspherical by the random forest model, as pointed out by the reviewer.

This apparent misclassification results from imperfect training data sets, as stated on P17L7-18 (initial manuscript), and can be understood upon consideration of Fig. S13 and S14 of our SI. From Fig. S13a and b it becomes clear that there exists aspherical particles with low TBC values, comparable with those of spherical particles (overlap of blue and red curves) within the (entire) calibration data set.

During the random sampling of (200'000) aspherical particles for the training of the random forest model (see P12L17-25, initial manuscript) aspherical particles with any TBC values described by the red curves in Fig. S13a and b can be selected. Hence, there is no a priori constrain to high TBC values for aspherical particles during the training of the random forest model, which is then trained to classify any such particle with the flag "isAspherical". In other

words, the misclassification results due the overlap of the TBC distributions of spherical and aspherical particles shown in Fig. S13a and b. This overlap in turn results mainly from the VOAG produced NaCl particles that show symmetrical scattering patterns thus associated with low TBC values (Fig. S16) and to a lesser extent also, from VOAG produced PEG particles which could exhibit non-symmetrical scattering patterns and hence relatively larger TBC values (Figs. S15) despite being spherical. Here, we have simply assigned any PEG and NaCl particle produced by the VOAG, as spherical and aspherical, respectively. An improved particle classification, also for the particles questioned by the reviewer, could be achieved through visual inspection of all particles/scattering patterns within the calibration data set and manual classification as either noise, spherical or aspherical, and then only train the random forest model on this cleaned data set (see P12L17-25, initial manuscript). However, this is not feasible for the large number of particles within our data sets, as explained in the text (P17L15-18).

We note that upon inspection of Fig. S13a and b that the majority of the aspherical NaCl particles clearly show higher TBC values compared to the spherical PEG particles. This results in the overall good and correct classification of particles by the random forest algorithm (see Fig. 6), by using a sufficiently large number of particles (in our case 200'000) for training, where statistically, the majority of aspherical particles encompasses TBC values different from those of the spherical particles.

We have now addressed this limitation more explicitly through various changes throughout the main text as indicated below:

Added the statement on P13L5 (revised manuscript):
*"Any PEG or NaCl particle produced as described in Sect. 3.1.1 and fulfilling these usability criteria, is defined as spherical and aspherical particle, respectively, within the calibration data set, without further visual inspection of the scattering pattern. However, it should be noted, that this approach does not filter out aspherical NaCl particles associated with rather symmetric scattering patterns and consequently low values for the symmetry parameters (see Fig. S16). Hence, this can explain the overlap in the distribution of for instance of the TBC values of both particles classes (see Fig. S13a and b) and thus potential misclassification of asymetrical NaCl as spherical particles."*

Added the statement on P13L19 (revised manuscript):
*"Selection of a sufficiently large number of aspherical particles for training the random forest model ensures that a statistical particle majority will show TBC values different from those observed for spherical particles (see Fig. S13a and b)."*

Added the statement on P15L15 (revised manuscript):
*"In addition, some of the misclassification can result from near-spherical NaCl particles within the training data set, as discussed above."*

Changed from P17L8 (initial manuscript):
*"Nevertheless, we note that there are particles classified as isAspherical, even though the scattering patterns appears symmetrical."*
To P18L12 (revised manuscript):
*"Nevertheless, we note that there are particles classified as isAspherical, even though the scattering patterns appears symmetrical, for instance, particles 39 and 46 in Fig. 8, which have values for the symmetry parameters comparable to particles classified as spherical (e.g. particle 29 in Fig. 8)."*

Added to P17L15 (initial manuscript), P18L22 (revised manuscript):
*"The consequence is that some particles become misclassified as e.g. aspherical, despite their overall symmetric scattering patterns (see above). This error could be reduced through manual visual inspection and manual selection and definition of particle class for every particle within the calibration data set, prior to training of the random forest model."*

Added to P17L18 (initial manuscript), P18L27 (revised manuscript):
*"The latter results from the majority of the spherical and aspherical particles within the calibration data set to distinctively differ in terms of their symmetry parameters (see Fig. S13)."*

Added to P25L27 (revised manuscript):
*"Finally, we have noted above that our random forest model is associated with a misclassification rate, resulting in some symmetrical scattering patterns to be classified as aspherical and vice versa (see Sect. 4.3 and Fig. 8). We have argued that this is a consequence of artifacts within the calibration data set (see SI Sect. S6.1) from which particles are randomly selected for the training of the classification algorithm. This error could be reduced and overall classification could be improved in future studies, upon manual cleaning of the calibration data set prior to model training."*

We have further changed SI Sect. S6.1 from (SI P15L12, initial SI):
*"Thus, even though the TBC in general is a good measure for particle sphericity, an absolute threshold value above which all particles are considered aspherical cannot be applied. This can partly explain the misclassification of clearly aspherical particles as spheres. Overall, this is a shortcoming of using the particle measures within the random forest model, rather than the individual pixel information."*
To SI P15L12 (revised SI):
*"Similarly, some of the NaCl particles produced by the VOAG reveal symmetrical scattering patterns, which are consequently associated with relatively low TBC values (see Fig. S16). While most of these near-spherical NaCl particles show a symmetric scattering pattern along only one of the CMOS arrays, we cannot exclude NaCl particles from our calibration data set that show symmetry comparable to spherical PEG particles, without manual inspection of these particles. Thus, even though the TBC in general is a good measure for particle sphericity, an absolute threshold value above which all particles are considered aspherical cannot be applied. Overall, this is a shortcoming of using the particle measures within the random forest model, rather than the individual pixel information, as well as defining all VOAG produced PEG and NaCl particles as spherical and aspherical, respectively, at the absence of a manual check of the individual scattering patterns."*

**P17 L34: Can you offer an explanation for the trimodal appearance of the AIC values for the 243K NH4NO3 case (Fig 10d)?**
We do not have a direct explanation for the trimodality of the AIC values, however, given that all the values are really low, i.e. below the threshold of 0.2 (as can be inferred from our Fig. S13d), we are confident these are all cloud droplets.

**SI P4 FigS4: It would be helpful to have *RTe* and *RAWe* defined in the caption (presumably real-time electronics board and raw sampling electronics?).**
We have now defined both terms in the figure caption.

**SI P6 FigS6: "*2 μm*" instead of "*2 mum*"**
We have corrected this in the figure caption now.

**SI TableS1: A minor suggestion, it would be helpful to have the data sets used in FigS12 highlighted by bold or coloured font.**

We have now highlighted the datasets used in Fig. S12 in bold and noted also that in the caption of Tab. S1.

Hussain, K., & Saunders, C. P. R. (1984). Ice nucleus measurement with a continuous flow chamber. *110*(463), 75-84. doi:10.1002/qj.49711046307

Lacher, L., Lohmann, U., Boose, Y., Zipori, A., Herrmann, E., Bukowiecki, N., . . . Kanji, Z. A. (2017). The Horizontal Ice Nucleation Chamber (HINC): INP measurements at conditions relevant for mixed-phase clouds at the High Altitude Research Station Jungfraujoch. *Atmospheric Chemistry and Physics, 17*(24), 15199-15224. doi:10.5194/acp-17-15199-2017

Mahrt, F., Marcolli, C., David, R. O., Grönquist, P., Barthazy Meier, E. J., Lohmann, U., & Kanji, Z. A. (2018). Ice nucleation abilities of soot particles determined with the Horizontal Ice Nucleation Chamber. *Atmos. Chem. Phys., 18*(18), 13363-13392. doi:10.5194/acp-18-13363-2018

Rogers, D. C. (1988). Development of a continuous flow thermal gradient diffusion chamber for ice nucleation studies. *Atmospheric Research, 22*(2), 149-181. doi:http://dx.doi.org/10.1016/0169-8095(88)90005-1

---

## Author Comment (AC2) · 23 May 2019

Reviewer comments are reproduced in **bold** and our responses in normal typeface; extracts from the originally submitted manuscript are presented in *red italic*, and from the revised manuscript in *blue italic*.

We have numbered the reviewer's major comments for ease of cross-reference within the other reviews.

**The authors present a new instrument, a High Speed Particle Phase Discriminator (PPD-HS), for phase discrimination of cloud particles. The instrument is an extension to the Small Ice Detector (SID) family but compared to the SID-3 and the Particle Phase Discriminator (PPD) it uses two CMOS arrays instead of a CCD camera. This modification allows recording of two 1D rows of scattering information, which reduces the amount of recorded data and allows higher detection rates of several hundred particle per second. The 1D scattering patterns are analyzed for their symmetry to discriminate between spherical and aspherical particles. The authors present characterization of the new instrument and have developed a supervise machine learning algorithm to automatically classify particles for their phase. The paper is well written and the used approach justified. However, few aspects should be addressed before publication.**

We thank the reviewer for carefully reading the manuscript and the overall constructive comments on it. We hope that the responses below satisfactorily address the reviewer concerns.

**Major comments:**
1. **The random forest model was trained with a test particle dataset of droplets and NaCl particles. However, it is somewhat questionable if NaCl particles would be a good proxy for aspherical ice particles. Why the random forest model was not trained using real ice particles from the experiment where NH4NO3 aerosol was frozen at T = 223 K?**
   The reviewer raises a valid point here, that real ice particles could have been used to train the random forest model. However, there are multiple reasons, why we decided to use VOAG data sets only for calibration purposes, as outlined below.
   The particle classification by the random forest model is purely based on particle shape analysis or more precisely the symmetry of the scattering pattern (P6L25-28, initial manuscript).The NaCl particles produced by the VOAG are aspherical, mimicking the aspherical features similar to that of ice crystals that is needed to test the working principle of PPD-HS. The reason for not using the ice data to train the model is to have independent data sets that can be used to test the performance of the trained algorithm. In fact, the overall good classification of particles in Figs. 7 and 10 provide direct evidence that the trained the random forest using both PEG and NaCl particles produced by the VOAG can successfully be applied to distinguish cloud droplets from ice crystals produced in HINC.

   Next, it should be noted that formation of ice crystals in HINC is limited to crystals with diameters < 10 μm, as we note on P10L3 (initial manuscript, see also comment #3 by reviewer#2). In order to test the performance of PPD-HS over a larger particle size range, for potential future use of PPD-HS in field measurements, where larger crystals can form,

we decided to use the VOAG to produce larger aspherical particles. Production of larger particles within HINC is limited by the residence time within the chamber as well as the gravitational settling of particles owing to the horizontal alignment of the chamber, which is circumvented in the VOAG setup shown in Fig. 3. A unique feature of the VOAG is the production of almost monomodal particle populations, which cannot be achieved to that extent in a CFDC setup as shown by Garimella et al. (2017), so that the random forest model would be mis-trained by the presence of e.g. small spherical ice particles (see Fig. 7). Finally, using the HINC data for training the random forest model would further be complicated by ice crystals and cloud droplets forming at different temperatures and relative humidity conditions that could influence the particle shape and hence bias the random forest model. By using the VOAG to produce both sets of spherical and aspherical particles we minimize such instrumental bias from the particle production mechanism, which is essentially the same for both particle types.

2. **Alternatively, would it have been possible to use SID-3 and PPD 3D scattering patterns of ice particles as training sets?**
The diffraction patterns captured e.g. by PPD-2K (see Fig. 1 in Vochezer et al. (2016)) or those shown in our Fig. 2a-c are post-processed to determine particle shape and are not _directly_ comparable to the scattering intensities captures by our linear CMOS arrays. One would need to transform the PPD-2K diffraction patterns captured by the CCD camera to scattering data along the two linear array stripes, as schematically depicted by our Fig. 2a. Since the goal of the present work was to characterize particle sphericity based on the linear scattering output of the newly presented PPD-HS (linear arrays allow for the high sampling rate),  and the optical geometry used within the new instrument, the usage of ice particle scattering patterns of PPD-2K or SID-3 would not contribute to the overall goal of the presented work and was therefore not done.

3. **The authors do not discuss what is the upper size limit of the PPD-HS.**
Please note that the largest particle size we have experimentally tested is 32 µm. It should further be noted that with the current experimental setup, we can only experimentally verify and cross-check particle sizing by PPD-HS for particles < 20 µm, i.e. those that lie within the size range of the APS (see P12L32-P13L1, initial manuscript) and for larger particles can only compare the instrument response of PPD-HS to theoretical calculations (see Fig. 5).
However, the limiting factor determining the upper size limit of PPD-HS is the focus size of the trigger laser beam and the response of the photo diode. We state that the trigger laser beam is focused to a depth of 100 µm at the plane, where particle and trigger laser beam intersect (see P4L29-21 in initial manuscript). Any particle needs to be contained within the beam in order to cause a trigger and become recorded later. Hence, the maximum particle size that can be detected is < 100 µm.
We have added the following statement to P5L4 (revised manuscript):
"_The focal depth of the trigger beam of 100 µm constrains the size of particles that can be detected by PPD-HS to this diameter, as each particle needs to be contained within the focus of the trigger laser beam in order to cause a trigger and become recorded later._"

Furthermore, the upper detection limit of PPD-HS is constrained by saturation of the detector for particles > 70 µm. Please see our answer to comment P6L11-12 of reviewer#1.

4. **Although the method is calibrated up to 32 µm using the test particles, the HINC experiments do not produce particles >10 µm. It is unclear how well the PPD-HS discriminates phase in the size range of 10-30 µm.**
It is true, that for the classification results depicted in Figs. 7, 10 and 11, we are limited to the particle sizes formed within HINC. However, we believe that classification in the range 10-30 µm can be achieved quantifiably and with sufficient accuracy. As shown in Fig. 3,

using the VOAG setup, we generated both spherical and aspherical particles up to 32 µm in size, as stated on P10L1-4 (initial manuscript). This data is also shown in Fig. 4 and a detailed overview of the total number of particles sampled for each type and size is given in our Tab. S1 in the SI. It is further stated in the initial manuscript (P12L21-25), that out of this entire pool of particle data and covering the entire size range, 400'000 particles were randomly chosen for training of the random forest model, with the remainder data being used for assessing classification model performance. Therefore, the confusion matrix presented in Fig. 6 can directly be used to assess the shape (and phase) discrimination capability of the random forest model for PPD-HS particle data covering the entire size range 3-32 µm. In order to make this more explicit, we have adapted the statement on P12L21-25 (initial manuscript) from:

*"Using these four predictors, we trained a random forest model on 400,000 randomly selected (training part), constituting of equal fractions of spherical and aspherical particles and using a total number of 200 trees (see SI Sect. S7). Classification performance was then tested on the remaining particle data (test data set; 4,371,162 particles), and subsequently applied to simulated hydrometeors from our HINC experiments."*

To read P13L16 (revised manuscript):

*"Using these four predictors, we trained a random forest model on 400,000 randomly selected (training part), constituting of equal fractions of spherical and aspherical particles and using a total number of 200 trees (see SI Sect. S7). These particles were randomly selected from the entire VOAG data set (see Tab. S1) covering the entire particle size range of 3-32 µm. Classification performance was then tested on the remaining particle data (test data set; 4,371,162 particles; same size range), and subsequently applied to simulated hydrometeors from our HINC experiments, where particle size is usually constrained to diameters < 10 µm."*

We have further changed the sentence P13L11 (initial manuscript) from:
*"In Fig. 6 we provide the classification results, when the trained random forest model is applied to the test data."*
To P14L12 (revised manuscript):
*"In Fig. 6 we provide the classification results, when the trained random forest model is applied to the test data, encompassing both PEG and NaCl particles of the sizes between 3-32 µm in diameter (see Tab S1)."*

A similar statement was added to the caption of Fig. 6 to read:
*"Confusion matrix of the random forest model applied to test data, i.e. the fraction of particles not used for model training, encompassing both spherical and aspherical particles of diameters between approximately 3-32 µm."*

5. **For example, how would 1D patterns of large droplets with multiple rings be classified? Also, larger complex ice particles have 3D scattering patterns showing frequent speckles (as seen in Fig. 2b). How would the symmetry of such particles look like in the PPD-HS?**
The 2D scattering pattern of a droplet with multiple diffraction rings, would result in a 1D scattering pattern captured by our linear CMOS arrays, similar to the one shown in e.g. Fig. S15 #3, hence still show symmetry in terms of TBC and/or AIC. Thus, as long as the diffraction pattern of a spherical particle yields a symmetric scattering pattern along the CMOS arrays, it would be correctly classified by our random forest model as spherical particle, since the random forest model just assesses the symmetry in terms of TBC and AIC, which are independent of the number of diffraction fringes. A similar reasoning applies to speckled ice particles and an example of such a particle when detected by PPD-HS is given in Fig. 2e.

6. **The dead time of the instrument (177 to 267 µs) is high and its consequences to sampling statistics are only discussed quantitatively in the supplementary materials. However, this discussion should be incorporated into the main text. In chapter**

**5 the issue is mentioned but there is no discussion of the implications for future ice spectrometer and field measurements. If at a typical mixed-phase concentration of 100 cm$^{-3}$ 100% or more particles are missed, how representative are the retrieved PPD-HS ice concentrations, especially if looking for the first ice?**

We agree with the comment of the reviewer, that the dead time of the CMOS arrays used within PPD-HS, which is the detection rate limiting factor, is high (T > 177 μs) compared to other optical particle instruments, such as SID-2H or PPD-2K with electronic dead times of T = 50.0 μs and T = 8.25 μs, respectively, (see e.g. Vochezer et al., 2016).

In fact, incorporation of this discussion into the main text has been part of extensive discussions among the authors during manuscript preparation. However, we believe that such discussion in the main text removes the focus from our main point, which is that reduced light scattering data captured by the two linear CMOS arrays can successfully be used for robust particle shape analysis and phase discrimination. It is true, though; that an upgrade to faster CMOS arrays with reduced dead times in future instruments should be considered when aiming at sampling atmospheric MPC. Certainly, the usage of the current CMOS arrays in PPD-HS with relatively large dead time is a tradeoff between instrument costs and frame rate. We hope to sufficiently address this aspect in the main text through the following changes.

We have added the following statement to P7L13 (revised manuscript):
*"We note that the minimum dead time of PPD-HS ( T = 177 μs) is relatively high, compared to that of SID-2H (T = 50.0 μs, Johnson et al. 2014) and PPD-2K (T = 8.25 μs, Vochezer et al., 2016)and consequently the fraction of missed particles at high flow rates and particle number concentrations. However, we note that adjustment of the total flow rate as well as of the integration delay and duration parameters determining the overall CMOS dead time is a unique property of PPD-HS, which can be used to empirically optimize the sampling conditions."*

We have also added the following statement to P25L10 (revised manuscript):
*"Furthermore, we have noted in Sect. 2.4 that the CMOS dead time of PPD-HS is relatively high, compared to similar devices. This causes the fraction of missed particles to be relatively high when sampling with PPD-HS at high particle number concentrations and using high total flow rates with consequences for sampling atmospheric MPC, where the ice fraction is (initially) low. Hence, upgrades to CMOS arrays with reduced dead time would be meaningful in view of potential future field applications of similar devices. Nevertheless, such changes do not affect the capability of using the reduced CMOS array scattering data to successfully determine particle shape."*

7. **Consequently, the last sentence in the abstract should be modified accordingly.**

We have changed the sentence from P1L18-20 (initial manuscript):
*"We conclude that PPD-HS constitutes a powerful new instrument to size and discriminate phase of cloud hydrometeors and thus study microphysical properties of mixed-phase clouds, that represent a major source of uncertainty in aerosol indirect effect for future climate projections."*

To P1L18 (revised manuscript):
*"From our laboratory experiments we conclude that PPD-HS constitutes a powerful new instrument to size and discriminate phase of cloud hydrometeors. The working principle of PPD-HS forms a basis for future instruments to study microphysical properties of atmospheric mixed-phase clouds, that represent a major source of uncertainty in aerosol indirect effect for future climate projections."*

**Minor comments**
**p.2, line 18: "This is despite the knowledge that cloud particle size distributions comprised of a mixture of cloud droplets and ice crystals are affected by the presence of small ice particles". Please provide references.**

We have now added the following references: Lawson, Baker, Schmitt, and Jensen (2001), Korolev, Isaac, Cober, Strapp, and Hallett (2003).

**p.2, line 34: Spherical particles can change the incident polarization state. The incident polarization state remains unchanged in the case it is linearly polarized (horizontal or vertical). This is why depolarization techniques use linearly polarized light.**

Thank you for spotting this. We agree with the reviewer, that when considering an incident _linearly_ polarized light, as in the case of PPD-HS, spherical particles do not cause depolarization of the _scattered_ light. We have formulated this more carefully now:

Changed from P2L34 (initial manuscript)

_"[…] making use of the fact that aspherical particles change the polarization of incident light, whereas spherical particles do not."_

To P2L34 (revised manuscript)

_"[…] making use of the fact that incident linearly polarized light is depolarized by aspherical particles, whereas spherical particles do not cause depolarization in the scattered intensity (Bohren & Huffman, 1983; Liou & Lahore, 1974)."_

**p.3, line 1: Depolarization techniques are sensitive to even small changes from a spherical shape, e.g. spheres with surface roughness or oblate/prolate particles do cause a measurable depolarization signal. However, commercial polarization sensors, like the one used in Zenker et al., 2017, can have difficulties discriminating small ice particles from droplets. These two techniques should not be mixed.**

We agree with the reviewer and have adapted the text accordingly.

From P3L1 (initial manuscript)

_"In the cases where small near-spherical ice crystals can form, depolarization techniques might be limited in the discrimination between spherical liquid drops and ice crystals."_

To P3L1 (revised manuscript)

_"In the cases where small near-spherical ice crystals form, commercial depolarization sensors frequently used for cloud composition analysis, might be limited in detecting the depolarization and thus the discrimination between liquid cloud droplets (causing no depolarization) and ice crystals (causing depolarization)."_

**p.3, line 3: The ability to discriminate phase based on particle shape depends on the used method. Shadow imaging and optical microscopy in best case have an optical resolution around 2 μm and, thus, cannot be used to discriminate the phase of small (<50 μm) cloud particles. 2D diffraction patterns can reveal more details that cannot be resolved by traditional imaging methods.**

We acknowledge the concern raised be the reviewer. Certainly, particle type discrimination based on particle shape is ultimately controlled by the optical resolution of the instrument. We have tried to clarify this aspect and reformulated our statement more carefully, which is intended to cover instruments using 2D diffraction patterns of cloud particles.

Changed from P3L3 (initial manuscript)

_"Particle shape, for instance, constitutes a powerful parameter that can be used to discriminate hydrometeor types (Hirst and Kaye, 1996)."_

To P3L5 (revised manuscript)

_"Particle shape, for instance, constitutes a powerful parameter that can be used to discriminate hydrometeor types, given a sufficiently high optical resolution of the instrument used for imaging. For instance, Hirst and Kaye (1996) have shown that analysis of 2D scattering profiles can be used to determine particle shape, which can then be related to particle phase."_

**p.3, line 9: small-scale complexity, please add the following references: Ulanowski et al., 2014; Schnaiter et al., 2016**

Thank you, we have added the suggested references.

**p.7, line 4: The discussion of the electronic dead time is crucial for understanding the instrument performance and should be added to the main text. The dead time of 177 to**

**267 µs is high compared to other cloud instrumentation, which will lead to reduced sampling volume. Fig. S11 shows that at a typical mixed-phase concentration of 100 cm⁻³ 100% or more particles are missed, which will have severe consequences for detecting ice in mixed-phase conditions.**
Please see our answer and changes to comment #6.

**Fig. 2: The example droplet (a) is a larger droplet with multiple visible rings that does not correspond with the scattering data from CMOS array (d) that shows 1 or possibly 2 maximums corresponding to a small droplet. How would scattering data from a larger droplet with multiple rings look like?**
A droplet with multiple rings would result in multiple peaks along each CMOS array at the location of these diffraction fringes (rings) overlap with the CMOS pixels. Example scattering patterns of such spherical particles can be found in our Figs. S20 and S21 and we have made a note of that in the caption of Fig. 2 in the revised manuscript.

**p.11, line 20: Are the calibration datasets from the VOAG experiments representative for cloud particles? First, the particle sizes are limited to 32 µm, whereas ice crystals can be significantly larger. More importantly, are salt particles a good proxy for ice crystal shapes? Why not use HINC experiments in cirrus conditions as training data sets?**
We agree that atmospheric cloud particles and ice crystals in particular can be much larger than the largest size tested here. However, the goal of the present study is to test and characterize the performance of PPD-HS for use as detector in a CFDC setup (as carefully noted in the title of the manuscript), where particle size is usually constrained to < 10 µm (P10L1-3 in the initial manuscirpt). We agree that it would be useful to test PPD-HS on larger particles of both spherical and aspherical shape prior to using it in e.g. field measurements, which is beyond the scope of this study. Furthermore, please see our answer to comment #1 above.
In order to clarify why the VOAG datasets were used for training of the random forest model, we have added the following statement to the text on P12L5 (revised manuscript):
*"Training of the random forest model was constrained to the PEG and NaCl particles generated by the VOAG, which we assume to have similar shape properties to spherical cloud droplets and aspherical ice crystals, respectively. Moreover, the vertical alignment of the VOAG setup allowed us to cover a larger size range without changing the experimental conditions, than would have been possible within the horizontal CFDC setup. Finally, by excluding the HINC-PPD-HS data from model training, these data sets provide independent data for testing classification model performance."*

**p.13, lines 4-5: The scattering cross section of aspherical particles between 10.6° to 101.0° can be very different to NaCl. Therefore, it cannot be stated that PPD-HS correctly sizes all aspherical particles <20 µm.**
The reviewer is right in pointing out that a randomly oriented particle of any shape might not have the same scattering cross section as the NaCl particles investigated here. However, the wide collection angle of the elliptical mirror results in only minor differences in sizing spherical and aspherical particles, such as hexagonal ice crystals (J. Ulanowski, personal communication). Furthermore, it should be noted the assumption of an idealized particle geometry is a limitation of any optical particle counter/instrument. Moreover, changes in detector sensitivity or variation of the trigger laser beam intensity are other parameters that can cause erroneous particle sizing. However, discussion of this is beyond the scope of the presented manuscript.
We have therefore tuned down our formulation on P13L4-5 (initial manuscript) from:
*"From these measurements we conclude that PPD-HS correctly sizes particles in the APS size range up to approximately 20 µm."*
To now read P14L2 (revised manuscript):
*"From these measurements we conclude that PPD-HS sizes particles in the range up to approximately 20 µm, where we can compare to our APS measurements, with reasonable accuracy."*

**Fig. 4: Please explain do,g and da,g in the figure caption.**

The caption of Fig. 4 already explains the parameter space in words. However, we adapted the caption slightly to make this more explicit:

*"[…] showing the geometric mean of the optical diameter ($d_{o,g}$) determined by PPD-HS as a function of the geometric mean of the aerodynamic diameter ($d_{a,g}$) obtained from the APS, […]"*

**Fig. 7: All the particle concentrations are given as counts and not as number concentrations within a volume. Since the sensitive area of the PPD-HS is known the instrument counts can be converted to concentrations. Also, it would be illustrative if the mean modal diameter would be marked in panel e.**

We feel that reporting particle detection in terms of counts per time is a more useful and intuitive quantity to assess the framerate of PPD-HS, so that volume-based number concentration would not be an improvement for our purpose. Hence, no change has been made.

It is true that a volume-based number concentration can be calculated under the consideration of the sensitive volume of PPD-HS and should be done when for instance comparing the ambient particle concentration to that detected by PPD-HS and/or comparing the detection rates of multiple instruments.

Similarly, we decided not to include the mean modal diameter as the mode will evolve as it is a function of RH (at a fixed T), as can be seen from our Fig. S19 and the discussion thereof.

**P. 24, line 8: What would be the maximum data acquisition rate of the RT-electronics if using priori specified parameters?**

In case the RT-electronics are operated independently from the RAW-electronics particle detection rates could reach approximately 3'000 particles per second. We have changed the statement on P24L8 in the initial manuscript from:

*"[…] whereas the RT-electronics have the benefit that they could (theoretically) achieve higher particle detection rates than presented here, but […]"*

To P25L8 (revised manuscript):

*"[…] whereas the RT-electronics have the benefit that they could (theoretically) achieve higher particle detection rates than presented here of approximately 3000 particles per second, but […]"*

**P. 25, line 12: Please define small.**

We note on P17L20-23 (initial manuscript), that the majority of the particles becomes correctly classified as aspherical ice crystals for sizes > 3.25 μm (see also SI Fig. S19) and further note d = 3 μm to be approximately the lower size limit for shape discrimination by PPD-HS (P24L14, initial manuscript).

*"Thus, small ice crystals with diameters below approximately 3 μm still remain a challenge for optical instruments such as PPD-HS."*

Bohren, C. F., & Huffman, D. H. (1983). *Absorption and scattering of light by small particles*. New York: Wiley Interscience.

Garimella, S., Rothenberg, D. A., Wolf, M. J., David, R. O., Kanji, Z. A., Wang, C., . . . Cziczo, D. J. (2017). Uncertainty in counting ice nucleating particles with continuous diffusion flow chambers. *Atmos. Chem. Phys. Discuss., 2017*, 1-28. doi:10.5194/acp-2016-1180

Johnson, A., Lasher-Trapp, S., Bansemer, A., Ulanowski, Z., & Heymsfield, A. J. (2014). Difficulties in Early Ice Detection with the Small Ice Detector-2 HIAPER (SID-2H) in Maritime Cumuli. *Journal of Atmospheric and Oceanic Technology, 31*(6), 1263-1275. doi:doi:10.1175/JTECH-D-13-00079.1

Korolev, A. V., Isaac, G. A., Cober, S. G., Strapp, J. W., & Hallett, J. (2003). Microphysical characterization of mixed-phase clouds. *Quarterly Journal of the Royal Meteorological Society, 129*(587), 39-65. doi:10.1256/gj.01.204

Lawson, R. P., Baker, B. A., Schmitt, C. G., & Jensen, T. L. (2001). An overview of microphysical properties of Arctic clouds observed in May and July 1998 during FIRE ACE. *106*(D14), 14989-15014. doi:10.1029/2000jd900789

Liou, K.-n., & Lahore, H. (1974). Laser Sensing of Cloud Composition: A Backscattered Depolarization Technique. *13*(2), 257-263. doi:10.1175/1520-0450(1974)013<0257:Lsocca>2.0.Co;2

Vochezer, P., Järvinen, E., Wagner, R., Kupiszewski, P., Leisner, T., & Schnaiter, M. (2016). In situ characterization of mixed phase clouds using the Small Ice Detector and the Particle Phase Discriminator. *Atmos. Meas. Tech., 9*(1), 159-177. doi:10.5194/amt-9-159-2016